# UniRec: Unified Multimodal Encoding for LLM-Based Recommendations

**Zijie Lei**                                                    *zijielei303@meta.com*
*Meta Monetization AI*

**Tao Feng**                                                     *taofeng2@illinois.edu*
*University of Illinois Urbana-Champaign*

**Zhigang Hua**                                                  *zhua@meta.com*
*Meta Monetization AI*

**Yan Xie**                                                      *yanxie@meta.com*
*Meta Monetization AI*

**Guanyu Lin**                                                   *guanyul@andrew.cmu.edu*
*Carnegie Mellon University*

**Shuang Yang**                                                  *shuangyang@meta.com*
*Meta Monetization AI*

**Ge Liu**                                                       *geliu@illinois.edu*
*University of Illinois Urbana-Champaign*

**Jiaxuan You**                                                  *jiaxuan@illinois.edu*
*University of Illinois Urbana-Champaign*

**Reviewed on OpenReview:** *https://openreview.net/forum?id=WXE255GWhQ*

## Abstract

Large language models (LLMs) have recently shown promise for multimodal recommendation, particularly with text and image inputs. Yet real-world recommendation signals extend far beyond these modalities. To reflect this, we formalize recommendation features into four modalities: text, images, categorical features, and numerical attributes, and emphasize unique challenges this heterogeneity poses for LLMs in understanding multimodal information. In particular, these challenges arise not only across modalities but also within them, as attributes (e.g., price, rating, time) may all be numeric yet carry distinct meanings. Beyond this intra-modality ambiguity, another major challenge is the nested structure of recommendation signals, where user histories are sequences of items, each carrying multiple attributes. To address these challenges, we propose `UniRec`, a unified multimodal encoder for LLM-based recommendation. `UniRec` first employs modality-specific encoders to produce consistent embeddings across heterogeneous signals. It then applies a triplet representation—comprising attribute name, type, and value—to separate schema from raw inputs and preserve semantic distinctions. Finally, a hierarchical Q-Former models the nested structure of user interactions while maintaining their layered organization. On multiple real-world benchmarks, `UniRec` outperforms state-of-the-art multimodal and LLM-based recommenders by up to 15%, while extensive ablation studies further validate the contributions of each component.

○ ulab-uiuc/UniRec    🤗 Hugging Face Collection

# 1 Introduction

Large language models (LLMs) have recently transformed recommender systems by reframing recommendation as a language modeling task (Geng et al., 2022; Li et al., 2023a; Zhang et al., 2023; Bao et al., 2023). Leveraging world knowledge and reasoning ability, LLM-based recommenders can capture rich semantic representations of users and items, enabling zero-shot prediction and explainable recommendations (Hou et al., 2023; Wang et al., 2025). However, most existing approaches primarily operate in text-centric settings, or at best combine text with images, where descriptive content such as reviews, metadata, or product visuals is abundant. While text and visual modalities are important, real-world recommendation data is far more heterogeneous, encompassing numerical, categorical, temporal, and geographical attributes that current LLM-based systems are ill-equipped to handle. Therefore, an open challenge remains: how can we design a unified framework that enables LLMs to effectively understand and reason over heterogeneous multimodal recommendation signals?

Addressing this challenge requires encoders that can faithfully represent such heterogeneous data. An effective encoder must be *schema-aware*, distinguishing attributes like price versus timestamp even when both are numeric; *hierarchy-aware*, capturing the nested structure of user histories as sequences of items with multiple attributes; and *modality-aware*, balancing signals across text, images, categorical fields, and numerical values. Naïve serialization into text or simple concatenation of embeddings discards these structural cues, obscures cross-feature interactions, and fails to capture sequential or relational patterns (Zhou et al., 2023a; Hou et al., 2023; Liu et al., 2023b; Singh et al., 2023; Bao et al., 2023).

While recent studies have begun exploring multimodal foundation models for recommendation (Geng et al., 2023; Luo et al., 2024), and vision–language models such as BLIP-2 demonstrate how Q-Formers can bridge modalities (Li et al., 2023b), these approaches are still limited in not tailoring to specific modality pairs or tasks and not providing a general solution for arbitrary heterogeneous recommendation inputs. **A unified framework is still missing**—one that can preserve schema, modality, and structural information while making heterogeneous signals accessible to LLM reasoning.

To fill this gap, we propose `UniRec`, a unified multimodal encoder that enables LLMs to leverage heterogeneous recommendation signals. `UniRec` employs modality-specific encoders to produce aligned embeddings for text, image, categorical, and numerical features. Each attribute is represented as a triplet—(attribute name, type, value)—to disentangle schema from raw inputs and preserve semantic distinctions. A hierarchical Query-Former then aggregates these representations, first into item-level embeddings and subsequently into user-level histories, explicitly maintaining the layered structure of interactions. Integrated with a pretrained LLM, `UniRec` enables reasoning over multimodal signals without losing structural context, thereby improving recommendation accuracy.

We validate `UniRec` on a suite of benchmark datasets spanning diverse recommendation scenarios with differing combinations of textual, visual, numerical, categorical, temporal, and geographical attributes. Across all settings, `UniRec` consistently outperforms state-of-the-art multimodal and LLM-based baselines and demonstrates robustness across datasets with varying attribute compositions, underscoring its effectiveness in both conventional and richly multimodal recommendation tasks.

# 2 Preliminaries

In this section, we first introduce the preliminaries of multimodal recommendation and formally define the problem setup. We then discuss the limitations of existing methods in this setting, which motivates the design of our proposed `UniRec` encoder.

## 2.1 Problem Setup

We begin by formalizing the task of multimodal sequential recommendation, where users interact with items over time and items are described by heterogeneous attributes across multiple modalities.

**Users and Items**. Let $\mathcal{U} = \{u_1, \ldots, u_{|\mathcal{U}|}\}$ be the set of users and $\mathcal{I} = \{i_1, \ldots, i_{|\mathcal{I}|}\}$ the set of items. Each user $u \in \mathcal{U}$ generates a chronological interaction history

$$\mathcal{H}_u = \big[(t_1, \ell_1, i_1), (t_2, \ell_2, i_2), \ldots, (t_T, \ell_T, i_T)\big] \tag{1}$$

where $t_j$ is the timestamp, $\ell_j$ an optional location, and $i_j \in \mathcal{I}$ the item at step $j$. The next-item prediction task is to model the conditional distribution

$$p(i_{T+1} \mid \mathcal{H}_u), \tag{2}$$

and use it to rank candidate items for recommendation.

**Attributes and Modalities**. Each item $i \in \mathcal{I}$ is described by a heterogeneous collection of attributes spanning multiple modalities, such as textual descriptions, images, categorical labels, numerical values, or spatiotemporal information. Formally, let $\mathcal{N}$ denote the attribute namespace and $\{\mathcal{V}_a\}_{a \in \mathcal{N}}$ the corresponding value domains. Then an item is associated with

$$\mathcal{A}(i) = \{(a, v) \mid a \in \mathcal{N}, \, v \in \mathcal{V}_a\}, \tag{3}$$

where $a$ is the attribute name and $v$ its observed value. Different attributes may share the same value format (e.g., both *price* and *rating* are numerical) yet carry distinct semantics.

## 2.2 Limitations of Existing Methods

Although prior work has made progress by leveraging textual and visual signals, existing approaches still face fundamental shortcomings when applied to heterogeneous multimodal recommendation.

**Loss of Schema and Structural Semantics**. Text-only LLM-based recommenders (Geng et al., 2022; Li et al., 2023a; Ye et al., 2024) flatten multimodal signals into text prompts. This process erases distinctions between different attribute roles and obscures the nested structure of interactions, making it difficult to preserve schema semantics or accurately model structured data.

**Shallow Fusion of Modalities**. Conventional multimodal recommenders (He & McAuley, 2016a; Wei et al., 2019b; Tao et al., 2022b) typically combine auxiliary signals such as images or reviews through simple concatenation or late fusion. Such shallow integration limits the model's ability to capture fine-grained cross-modal dependencies or hierarchical structures within user–item interactions.

**Limited Generalization Across Modalities**. Recent LLM–multimodal hybrids (Geng et al., 2023; Luo et al., 2024; Zhang et al., 2025a; López-Ávila & Du, 2025) demonstrate improved multimodal reasoning but are usually tailored to specific modality pairs and lack explicit schema-awareness. This reduces their generalizability across diverse recommendation settings with heterogeneous attribute types.

# 3 UniRec: Unified Multimodal Encoding for LLM-Based Recommendations

To address the limitations identified in the previous section, we propose `UniRec` for multimodal LLM-based recommendation. We first formalize heterogeneous signals into four modalities with modality-specific encoders to obtain reliable feature representations. On top of these, we design a schema-preserving triplet representation and a hierarchical Q-Former aggregation mechanism, enabling LLMs to effectively understand heterogeneous recommendation signals for next-item prediction. The details are introduced as follows.

## 3.1 Modality-Specific Encoders

Robust modality-wise encoders and dense, comparable representations are crucial for stable multimodal fusion (Li et al., 2024; Vouitsis et al., 2024; Xu et al., 2022). To this end, we map all inputs into 1024-dimensional embeddings using modality-specific encoders. For **text** (e.g., titles, reviews), we employ Qwen3-0.6B Embedding[1]. **Categorical labels** (e.g., product categories) are encoded with the same model using

---

[1] `https://huggingface.co/Qwen/Qwen3-Embedding-0.6B`

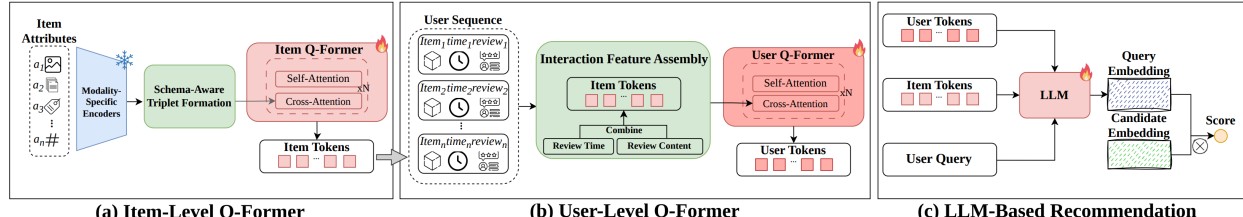

Figure 1: **UniRec Model Architecture:** (a) Item-Level Q-Former: Raw item attributes across heterogeneous modalities (text, categorical, image, numerical) are processed by modality-specific encoders and triplet formation. These generate schema-aware attribute embeddings, which are then aggregated by the Item Q-Former to produce a fixed-length item representation ($\mathbf{z}_t$). (b) User-Level Q-Former: A user's chronological interaction history, consisting of learned item tokens ($\mathbf{z}_t$), multimodal review contexts ($\mathbf{c}_t$), and timestamp embeddings ($\mathbf{p}_t$), is processed by an Interaction Feature Assembly module. The resulting sequence of combined interaction embeddings is then distilled by the User Q-Former into a unified user representation ($\mathbf{U}$). **The arrow passing the Learned Item Tokens from (a) to (b) explicitly models the nested structure of recommendation signals—where a user's history is a sequence of items, and each item is a collection of heterogeneous attributes.** (c) LLM-Based Recommendation: The learned user representation and item representations are projected as soft prompts to condition the LLM for next-item prediction, ranking against a corpus of candidate item embeddings.

category-aware instructions, avoiding the mixing of sparse one-hot features with dense vectors—a practice known to cause semantic misalignment and training instability (Guan et al., 2022; Li et al., 2022; Cheng et al., 2022). For **images**, we use CLIP ViT-B/32 model (Radford et al., 2021a)[2], followed by a projection layer that maps the native 512D representations to 1024D. For **numerical features**, we adopt a Fourier-based Math-Aware Number Encoder (Zhou et al., 2025; Cao et al., 2025), which integrates Fourier components (sine/cosine with log-spaced frequencies), raw magnitude/sign values, and a small learned projection. The encoder is trained with objectives enforcing additivity, invertibility, and distance preservation, while domain-specific adaptations handle temporal cycles (e.g., hour-of-day, month-of-year) and geospatial coordinates projected onto the unit sphere. Implementation details are provided in Appendix A.

## 3.2 Hierarchical Q-Former Encoder

Recommendation data has a natural nested structure: users interact with items, and each item carries multiple heterogeneous attributes. A flat architecture would conflate attribute-to-item and item-to-user aggregation. Our two-level design separates these concerns: the Item Q-Former compresses heterogeneous attributes into compact item tokens, while the User Q-Former captures sequential patterns across items. To model the nested structure of user-item interactions, we introduce a two-stage Hierarchical Q-Former. This architecture first distills an item's raw multimodal attributes into a fixed-size *item representation*, and then aggregates a sequence of item interactions into a final *user representation*.

**Schema-Aware Attribute Representation**. To preserve the semantic meaning of each attribute (e.g., knowing that "19.99" is a "price"), we represent it as a triplet of its ($name, type, value$). We obtain embeddings for the attribute's name ($\mathbf{a}_j$), its modality type ($\mathbf{t}_j$), and its value ($\mathbf{v}_j$). These are fused via summation into a single *schema-aware attribute embedding* $\mathbf{h}_j$:

$$\mathbf{h}_j = \mathbf{a}_j + \mathbf{t}_j + \mathbf{v}_j \tag{4}$$

The complete set of attribute embeddings for an item $i$ is denoted by $\mathbf{H}_i = \{\mathbf{h}_1, \ldots, \mathbf{h}_{N_i}\}$, which serves as the input for the first stage of our hierarchy.

This design resolves a fundamental ambiguity: a raw value like "4.5" could represent a price, a rating, or a weight. The triplet attaches what the value represents (name) and how it was encoded (type), analogous to column headers in a structured database.

---

[2]https://huggingface.co/sentence-transformers/clip-ViT-B-32

**Two-Stage Hierarchical Aggregation**. Our aggregation process uses two sequential Q-Formers.

First, an **Item Q-Former** processes the variable-length set of an item's attribute embeddings, $\mathbf{H}_{i_t}$. Using a set of learnable queries $\mathbf{Q}_{\text{item}}$, it distills this information into a fixed-length item representation, $\mathbf{z}_t$:

$$\mathbf{z}_t = \text{QFormer}_{\text{item}}(\mathbf{Q}_{\text{item}}, \mathbf{H}_{i_t}) \in \mathbb{R}^{K_{\text{item}} \times d} \tag{5}$$

Next, a **User Q-Former** aggregates the sequence of interactions over time. For each step $t$, the item representation $\mathbf{z}_t$ is combined with its associated *review context* embedding $\mathbf{c}_t$ and a *timestamp embedding* $\mathbf{p}_t$ to preserve chronological order. A new set of queries, $\mathbf{Q}_{\text{user}}$, processes this entire sequence to yield the final user representation $\mathbf{U}$:

$$\mathbf{U} = \text{QFormer}_{\text{user}}(\mathbf{Q}_{\text{user}}, \{\text{Concat}(\mathbf{z}_t, \mathbf{c}_t) + \mathbf{p}_t\}_{t=1}^{T}) \in \mathbb{R}^{K_{\text{user}} \times d} \tag{6}$$

This design yields rich, multi-token representations for both items ($\mathbf{z}_t$) and the user ($\mathbf{U}$), capturing more granular information than a single aggregated vector.

We denote the number of learnable query tokens as $N_q$. Each item's heterogeneous attributes are compressed into $N_q$ fixed-length representations; we analyze the sensitivity to $N_q$ in Section 4.3

### 3.3 UniRec Training and Inference

Our training strategy decouples representation learning from LLM adaptation in a two-stage process. First, we pretrain the `UniRec` encoder with a frozen LLM to learn aligned representations. Second, we fine-tune the encoder and the LLM jointly for the next-item prediction task.

**Pretraining Stage**. The objective of this stage is to train the modality-specific encoders and the Hierarchical Q-Former to produce a well-structured latent space while the LLM remains frozen. We employ a multi-task learning framework that combines two objectives. The first is a *reconstruction loss* ($\mathcal{L}_{\text{recon}}$), which ensures the Q-Former's output retains modality-specific details by using an MLP head to reconstruct the original attribute embeddings from the output tokens. The second is a *contrastive loss* ($\mathcal{L}_{\text{contrast}}$), which uses InfoNCE (van den Oord et al., 2018) to learn semantic item similarity by treating adjacent items in user histories as positive pairs. The final pretraining objective is a weighted sum of these two losses:

$$\mathcal{L}_{\text{pretrain}} = \mathcal{L}_{\text{contrast}} + \lambda_{\text{recon}}\mathcal{L}_{\text{recon}} \tag{7}$$

where $\lambda_{\text{recon}}$ is a balancing hyperparameter. This approach ensures the encoder learns representations that are both comprehensive and semantically aligned.

**Fine-Tuning Stage**. In the second stage, we adapt the system for recommendation by jointly training the Hierarchical Q-Former and the LLM's Low-Rank Adaptation (LoRA) weights (Hu et al., 2021), while keeping the core modality encoders frozen. The modality encoders (Qwen3-Embedding, CLIP ViT-B/32, MWNE) remain frozen throughout training to prevent overfitting on the relatively small recommendation datasets and to avoid catastrophic forgetting of pretrained representations, following the design principle of BLIP-2 (Li et al., 2023b). The final user representation is projected into the LLM's word embedding space, functioning as a *soft prompt* that conditions the LLM on the user's multimodal history.

The fine-tuning objective is the InfoNCE loss, applied to the next-item prediction task. The model learns to distinguish the ground-truth next item from a set of in-batch negative samples:

$$\mathcal{L}_{\text{finetune}} = -\log \frac{\exp(\text{sim}(\mathbf{u}, \mathbf{z}_{T+1})/\tau)}{\exp(\text{sim}(\mathbf{u}, \mathbf{z}_{T+1})/\tau) + \sum_{j=1}^{N} \exp(\text{sim}(\mathbf{u}, \mathbf{z}_j^{-})/\tau)} \tag{8}$$

Specifically, for a batch of $B$ user sequences, the ground-truth next item for each sequence serves as the positive, while the remaining $B-1$ target items in the batch serve as negative candidates, following the standard InfoNCE framework.

This joint training allows the Q-Former to refine its representations for the recommendation task while teaching the LLM to interpret the injected soft prompts.

Table 1: Detailed summarization of user and item attributes in Beauty, Baby and Yelp datasets.

| Dataset | Level | Attributes |
|---------|-------|-----------|
| Beauty | User | timestamp, rating, title, text, image |
|        | Item | main_category, title, average_rating, features, description, price, image, store, categories, details |
| Baby | User | timestamp, rating, title, text, image |
|      | Item | main_category, title, average_rating, features, description, price, image, store, categories, details |
| Yelp | User | review_date, review_text, review_star |
|      | Item | name, latitude, longitude, stars, review_count, attributes, categories, image, image_caption, image_label |

**Inference and Ranking**. During inference, a final user representation $\mathbf{u}$ is generated by processing the user's interaction history and mean-pooling the LLM's final hidden state. To produce a ranked list, this user embedding is used to compute a relevance score via dot product similarity against a corpus of pre-computed item embeddings, $s(u, i) = \mathbf{u} \cdot \mathbf{z}_i$. Candidate items are then ranked in descending order of this score.

## 4 Experiments

In this section, we conduct a comprehensive set of experiments on diverse and heterogeneous recommendation datasets to evaluate the performance of `UniRec`, and perform detailed ablation studies to validate the contribution of each design component.

### 4.1 Experiment Setup

**Tasks and Datasets**. Table 1 summarizes the attributes available in the datasets used for evaluation. We consider three benchmarks: the *Beauty* and *Baby* categories from the Amazon Product Reviews corpus (McAuley et al., 2015), and the Yelp Review dataset (Yelp Inc., 2018). Following common practice, we apply 5-core filtering to retain only users and items with at least five interactions. Each training sample is constructed by extracting 20 consecutive interactions as the historical sequence, with the 21st interaction designated as the ground-truth item. For evaluation, we adopt the leave-one-out strategy: the held-out ground-truth item is ranked against 99 randomly sampled negatives, yielding a candidate set of 100 items per user.

As shown in Table 1, user-side interactions include timestamps, ratings, and review text, with the Amazon datasets also containing review images. Item-side information covers textual descriptions, categorical labels, numerical values, and images. Amazon data thus provides multimodal signals at both user and item levels, combining product metadata with user-contributed visuals, while Yelp emphasizes spatiotemporal and categorical attributes, such as latitude–longitude coordinates, business categories, and review counts, alongside textual reviews and star ratings. This organization reflects the nested structure of recommendation signals, where each user history is a sequence of items, and each item is described by multiple heterogeneous attributes.

**Baselines and Metrics**. We evaluate a variety of baseline methods across three scenarios, grouped into **(a) Feature-based sequential recommenders**, **(b) Multimodal recommendation models**, and **(c) LLM-based multimodal recommenders**. For evaluation, we follow the next-item prediction setup with leave-one-out strategy: for each user, the ground-truth item is ranked against 99 randomly sampled nega-

Table 2: **Performance comparison on 3 datasets using MRR, Hit@10, and NDCG@10.** Baseline models are grouped into three categories: *multimodal-feature sequential*, *multimodal recommendation*, and *LLM-based multimodal*. **Bold** and underline denote best and second-best results. [†]Results reproduced by us using the official codebases.

| | Beauty | | | Baby | | | Yelp | | |
|---|---|---|---|---|---|---|---|---|---|
| Model | MRR | Hit@10 | NDCG@10 | MRR | Hit@10 | NDCG@10 | MRR | Hit@10 | NDCG@10 |
| *Multimodal-Feature Sequential Models* | | | | | | | | | |
| GRU4Rec | 0.2087 | 0.4215 | 0.2478 | 0.1365 | 0.2780 | 0.1532 | 0.4120 | 0.7815 | 0.4920 |
| BERT4Rec | 0.2215 | 0.4391 | 0.2605 | 0.1422 | 0.2919 | 0.1605 | 0.4182 | 0.7890 | 0.4975 |
| SASRec | 0.2549 | 0.4598 | 0.2890 | 0.1610 | 0.3387 | 0.1860 | 0.4335 | 0.8012 | 0.5128 |
| *Multimodal Recommendation Models* | | | | | | | | | |
| VBPR | 0.2476 | 0.4503 | 0.2818 | 0.1469 | 0.3413 | 0.1773 | 0.4605 | 0.8390 | 0.5433 |
| MMGCN | 0.3130 | 0.5045 | 0.3465 | 0.1848 | 0.3849 | 0.2170 | 0.5222 | 0.8769 | 0.6018 |
| BM3 | 0.3305 | 0.5628 | 0.3864 | 0.2166 | 0.4365 | 0.2501 | 0.5405 | 0.8912 | 0.6187 |
| LGMRec | 0.3433 | 0.5861 | 0.4025 | 0.2279 | 0.4522 | 0.2668 | 0.5530 | 0.9025 | 0.6261 |
| *LLM-based Multimodal Recommenders* | | | | | | | | | |
| IISAN | 0.2513 | 0.4492 | 0.2851 | 0.1476 | 0.3398 | 0.1784 | 0.4627 | 0.8426 | 0.5462 |
| MLLM-MSR | 0.3249 | 0.5713 | 0.3976 | 0.2084 | 0.4341 | 0.2495 | 0.5561 | 0.9014 | 0.6352 |
| MLLMRec[†] | 0.3581 | 0.6018 | 0.4213 | 0.2195 | 0.4352 | 0.2548 | 0.5382 | 0.8845 | 0.6168 |
| NoteLLM-2[†] | 0.3524 | 0.5955 | 0.4128 | 0.2095 | 0.4228 | 0.2452 | 0.5278 | 0.8742 | 0.6065 |
| **UniRec** | **0.3737** | **0.6270** | **0.4449** | **0.2635** | **0.4673** | **0.2977** | **0.5622** | **0.9150** | **0.6489** |

tives. Performance is reported using three standard top-*K* ranking metrics: *Mean Reciprocal Rank (MRR)* (Voorhees & Tice, 1999; Cremonesi et al., 2010), *Hit Rate at 10 (Hit@10)*, and *Normalized Discounted Cumulative Gain at 10 (NDCG@10)* (Järvelin & Kekäläinen, 2002; Burges et al., 2005; Liu, 2009), where higher values indicate better recommendation quality.

- **Multimodal-Feature Sequential Models.** This group adapts classical sequential recommenders by enriching item embeddings with multimodal features. Specifically, GRU4Rec (Hidasi et al., 2016), BERT4Rec (Sun et al., 2019), and SASRec (Kang & McAuley, 2018) are implemented using representations extracted from a pre-trained CLIP ViT-B/32 model (Radford et al., 2021a). CLIP encodes product text and images into a shared embedding space, which replaces ID embeddings while leaving the original model architectures unchanged.

- **Multimodal Recommendation Models.** These approaches are explicitly designed to integrate multimodal signals into recommendation. VBPR (He & McAuley, 2016a) augments matrix factorization with visual preference vectors derived from product images. MMGCN (Wei et al., 2019b) builds modality-specific user–item graphs and aggregates them with graph neural networks. BM3 (Zhou et al., 2023b) introduces a self-supervised learning framework that applies dropout to generate contrastive views without explicit negative sampling, improving robustness. LGMRec (Guo et al., 2024) decouples collaborative filtering and content modeling by combining local modality-specific graphs with global hypergraph-based embeddings, achieving stronger alignment between content and interaction signals.

- **LLM-based Multimodal Recommenders.** Recent methods leverage large language models to reason over multimodal content. IISAN (Fu et al., 2024) adopts a parameter-efficient fine-tuning framework that decouples intra- and inter-modal adaptation, improving both training efficiency and scalability while maintaining accuracy. MLLM-MSR (Ye et al., 2025) takes a summarization approach: it converts product images into textual descriptions, uses an LLM to summarize user histories, and fine-tunes the model for sequential recommendation. NoteLLM-2 (Zhang et al., 2025a) proposes a Multimodal Large Representation Model (MLRM) that leverages multimodal in-context learning and late fusion to generate robust dense representations from text and images. MLLMRec (Wang et al., 2025) takes a fundamentally different approach by directly prompting a native multimodal LLM to reason about user preferences, bypassing the encoder–bridge–LLM pipeline entirely, and integrates collaborative filtering graphs in its later version. These models highlight the emerging trend of directly coupling multimodal inputs with LLM.

**Implementation Details**. We implement `UniRec` on top of the Qwen3-Embedding-0.6B model with LoRA adaptation and integrated Q-Formers. All modalities are mapped into a unified 1024-dimensional space. Training updates Q-Former and LoRA parameters using the AdamW optimizer ($\beta_1 = 0.9$, $\beta_2 = 0.999$, learning rate $1 \times 10^{-4}$, weight decay 0.01) with linear warm-up (20 steps) followed by cosine decay. We employ the InfoNCE loss with temperature 0.07, training for 50 epochs with batch size 16 (accumulation step 1), and evaluate with batch size 32. LoRA is applied to attention and feed-forward layers with rank 16, $\alpha = 32$, and dropout 0.1. To stabilize and accelerate training, we enable FP16 precision, gradient clipping (norm 1.0), gradient checkpointing, and sequence length grouping. All experiments are conducted on a single NVIDIA A6000 GPU.

## 4.2 `UniRec` Outperforms State-of-the-Art Baselines

In the Beauty, Baby, and Yelp datasets, we compare `UniRec` against multimodal-feature sequential models, multimodal recommendation models, and recent LLM-based multimodal recommenders, as shown in Table 2. Across all three datasets, `UniRec` consistently outperforms the strongest baselines by relative margins of up to 16% in MRR, establishing a new state-of-the-art.

In the product recommendation setting on Beauty and Baby, where specialized multimodal models such as LGMRec dominate, `UniRec` delivers notable improvements—8.8% on Beauty and 15.6% on Baby in MRR—highlighting that its schema- and hierarchy-aware design provides clear advantages in modeling complex multimodal attributes. In the point-of-interest recommendation setting on Yelp, which emphasizes spatiotemporal and categorical diversity, `UniRec` further surpasses the strongest LLM-based competitor MLLM-MSR with a 1.1% gain in MRR, demonstrating robustness beyond product domains. Together, these results confirm that `UniRec` generalizes effectively across heterogeneous recommendation scenarios, ranging from multimodal product domains to geographic and categorical POI recommendation, consistently outperforming both classical multimodal architectures and modern LLM-based methods.

To validate that the sampled evaluation (99 negatives) does not distort our conclusions, we conducted full-catalog ranking on Beauty (all 12,101 items); the relative ordering of methods is preserved (UniRec MRR 0.0552 vs. LGMRec 0.0483, MLLM-MSR 0.0431), consistent with findings by Krichene & Rendle (2020). Full results are in Appendix G.1.

We additionally compare with MLLMRec (Wang et al., 2025) and NoteLLM-2 (Zhang et al., 2025a), which represent state-of-the-art alternative paradigms. On Beauty, UniRec outperforms both MLLMRec (MRR 0.3737 vs. 0.3581) and NoteLLM-2 (MRR 0.3737 vs. 0.3524) while using 7.7× fewer tokens per inference. The performance gap widens on Yelp, where geospatial features such as latitude and longitude are critical—text serialization destroys spatial proximity (e.g., "37.7749" and "37.7750" tokenize very differently), while UniRec's Fourier-based encoder preserves it by construction. A detailed comparison is in Appendix G.2.

UniRec's Q-Former compresses ~700 input tokens to 160, yielding a 2.8× inference speedup and 2.8× memory reduction versus text-only LLM baselines at batch size 64 on a single A100 GPU; a detailed efficiency analysis is in Section 4.4.

## 4.3 Ablation Studies Validate the Design of `UniRec`

We perform a comprehensive set of ablation studies to validate the effectiveness of `UniRec`'s design. By systematically removing or varying its core components, we isolate their individual contributions and assess how each choice impacts the overall performance.

**Schema- and Hierarchy-Aware Design Matters**. This ablation examines the importance of `UniRec`'s schema- and hierarchy-aware components: the triplet representation at the item level and the user-level summarization tokens at the sequence level. By selectively removing these elements, we isolate their individual and joint contributions to recommendation performance.

- **w/o Both.** Neither triplet representation nor user-level tokens are used, representing the minimal configuration.
- **w/o Triplet.** Removes triplet decomposition of item attributes, while retaining user-level tokens.

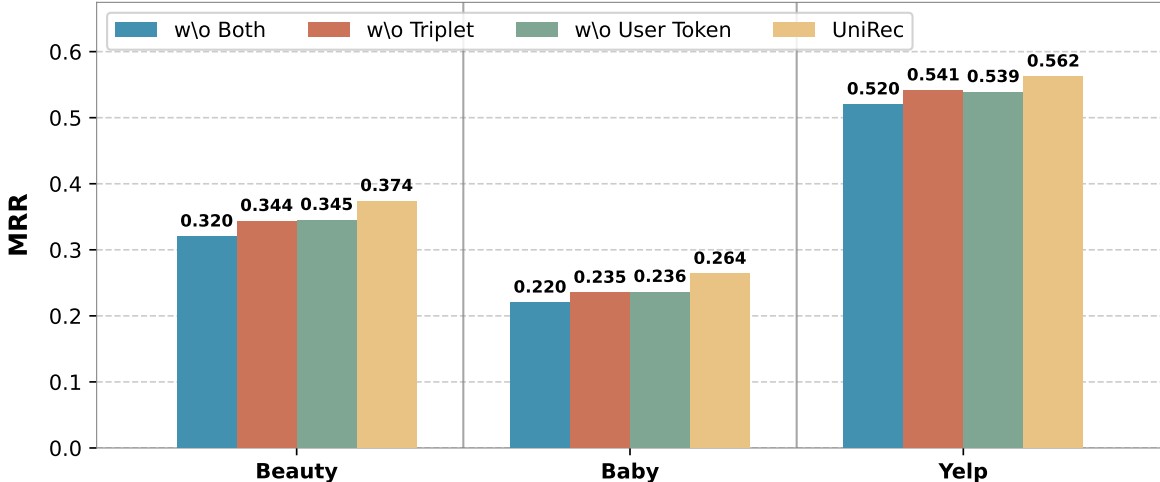

Figure 2: **Both schema- and hierarchy-aware components are crucial for `UniRec`'s performance.** Results are shown on Beauty, Baby, and Yelp datasets (measured in MRR). Performance improves step by step as components are added: starting from the minimal configuration (*w/o Both*), introducing either triplet representation or user-level tokens yields clear gains, while combining both achieves the highest performance.

- **w\o User Token.** Keeps triplet representation for items but discards user-level summarization tokens.

As shown in Figure 2, performance improves consistently as each component is introduced. Starting from the minimal setting (*w/o Both*), adding either triplet representation or user-level tokens yields notable gains, showing that each contributes independently. The best performance is achieved when both are enabled, confirming their complementary roles: triplet representation structures multimodal item attributes effectively, while user-level tokens capture nested dependencies across interaction histories. Together, these design choices allow `UniRec` to better exploit schema- and hierarchy-rich recommendation data.

**Query-Based Fusion Outperforms Alternatives**. We ablate the *fusion mechanism* used to combine modality-specific embeddings, with results reported in Table 3. All variants adopt the same Qwen3-0.6B embedding LLM as backbone and share identical candidate sets, modality-specific encoders, and training schedules. The only difference lies in the fusion design, for which we compare the following:

- **Pure Text.** Uses only item textual descriptions (e.g., titles/reviews) as content features, discarding other modalities. This text-only setup is a common baseline in multimodal recommendation studies (Zhou et al., 2023a; Liu et al., 2023a).
- **MLP Fusion.** Concatenates embeddings from all modalities into a single vector and processes them with a multilayer perceptron ("early fusion"), a simple but shallow cross-modal strategy (Baltrušaitis et al., 2019).
- **CLIP-Style Projection.** Maps each modality into a shared latent space through modality-specific linear layers, with alignment guided by a contrastive objective, following dual-encoder vision–language paradigms such as CLIP and ALIGN (Radford et al., 2021b; Jia et al., 2021).
- **Self-Attention Fusion.** Treats modality embeddings as tokens and applies a Transformer-style self-attention layer to capture pairwise interactions (Vaswani et al., 2017).

The results show that *Pure Text* already forms a strong baseline, highlighting the informativeness of textual signals. *MLP Fusion* slightly underperforms, suggesting that naive concatenation introduces noise without modeling schema distinctions. *CLIP-Style Projection* offers modest gains by better aligning text and image, yet its design is limited to pairwise modality alignment and struggles with heterogeneous attributes. *Self-Attention Fusion* achieves stronger results, confirming the value of richer cross-modal interactions, but still fails to capture hierarchical user–item structures. In contrast, `UniRec` consistently surpasses all alternatives

Table 3: **Query-based hierarchical fusion achieves the strongest results.** Ablation on fusion mechanisms across Beauty, Baby, and Yelp datasets. All models share identical encoders, training schedules, and candidate sets; only the *fusion strategy* differs. **Bold** and underline denote the best and second-best results.

| Fusion Mechanism | Beauty | | | Baby | | | Yelp | | |
|---|---|---|---|---|---|---|---|---|---|
| | MRR | Hit@10 | NDCG@10 | MRR | Hit@10 | NDCG@10 | MRR | Hit@10 | NDCG@10 |
| Pure Text | 0.3025 | 0.5016 | 0.3358 | 0.1857 | 0.3941 | 0.2179 | 0.4870 | 0.8643 | 0.5710 |
| MLP | 0.2980 | 0.4950 | 0.3301 | 0.1830 | 0.3865 | 0.2135 | 0.4825 | 0.8570 | 0.5655 |
| CLIP | 0.3150 | 0.5205 | 0.3482 | 0.1915 | 0.4040 | 0.2230 | 0.4962 | 0.8725 | 0.5795 |
| Self-attention | 0.3330 | 0.5635 | 0.3920 | 0.2140 | 0.4365 | 0.2520 | 0.5288 | 0.8955 | 0.6120 |
| Eq.(4): Concat+MLP | 0.3729 | 0.6285 | 0.4438 | 0.2628 | 0.4660 | 0.2965 | 0.5615 | 0.9138 | 0.6475 |
| Eq.(4): Gated | 0.3748 | 0.6278 | 0.4455 | 0.2641 | 0.4685 | 0.2982 | 0.5630 | 0.9155 | 0.6495 |
| `UniRec` | **0.3737** | **0.6270** | **0.4449** | **0.2635** | **0.4673** | **0.2977** | **0.5622** | **0.9150** | **0.6489** |

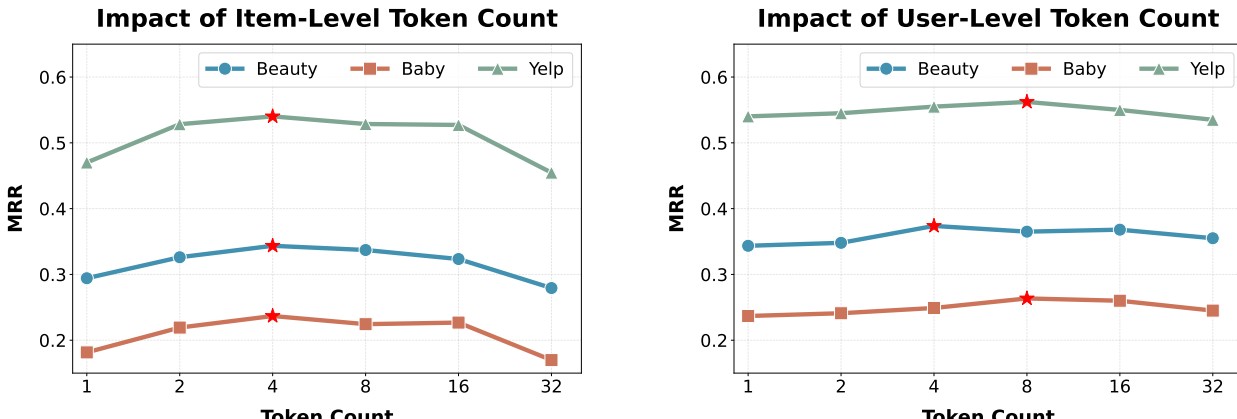

Figure 3: **Optimal token counts emerge for both item- and user-level Q-Formers.** Left: item-level tokens. Right: user-level tokens. Each curve shows MRR on one dataset, and the red star ($\star$) marks the token count achieving the highest performance.

across datasets and metrics, demonstrating that schema-aware, hierarchical query-based fusion provides a principled and robust solution for multimodal recommendation.

We additionally ablate the triplet fusion operator in Eq. (4), comparing addition (our default) against Concat+MLP (8.4M added parameters) and Gated fusion (9.4M added parameters) on Beauty. As shown in Table 3, all three achieve virtually identical end-to-end performance (MRR within 0.002, Hit@10 within 0.002). This is because the triplet components are empirically near-orthogonal in 1024-dimensional space (mean $|\cos\theta| \approx 0.013$–$0.022$), making addition a near-lossless superposition with zero added parameters and $71$–$100\times$ lower fusion latency than the learned alternatives.

**Token Count Sensitivity Reveals a Sweet Spot**. We examine how the number of latent tokens in the item-level and user-level Q-Formers affects performance (Figure 3). The results show that token count is crucial for balancing expressiveness and generalization. At the item level, accuracy peaks at 4 tokens, after which additional tokens yield diminishing or even negative returns, suggesting that a compact set of latent tokens suffices to capture key multimodal attributes. In contrast, the user-level Q-Former benefits from a slightly larger capacity, with optimal performance around 4–8 tokens, reflecting the greater complexity of modeling long interaction histories.

Overall, the trends reveal a trade-off: too few tokens underfit the data, while too many introduce redundancy, overfitting, or unstable training. These findings highlight the importance of careful token calibration at both item and user levels to ensure robust hierarchical representation learning.

**Modality Combination Ablation**.    To quantify the contribution of each modality group, we measure the cosine similarity and rank correlation between the full representation and representations obtained by systematically dropping modality groups:

Table 4: Impact of removing each modality group on item representation similarity.

| Configuration | Fields Kept | Cos Sim | Rank Corr |
|---|---|---|---|
| Full (all modalities) | 14 | 1.000 | 1.000 |
| Drop image | 13 | 0.973 | 0.877 |
| Drop numerical | 11 | 0.956 | 0.838 |
| Drop text | 11 | 0.949 | 0.793 |
| Drop category | 7 | 0.932 | 0.594 |
| Text only | 3 | 0.803 | 0.352 |

Each modality contributes incrementally, with categorical features providing the largest individual contribution (dropping from 14 to 7 fields), followed by text, numerical, and image. The degradation is graceful and monotonic—no single modality removal causes catastrophic failure, confirming UniRec's robustness to missing modalities.

We also compared the degradation behavior of addition versus Concat+MLP under missing modalities. When dropping the text modality (3 fields), additive fusion achieves $\text{cos\_sim}(\text{full}, \text{partial}) = 0.949$, while Concat+MLP achieves only 0.912. This is because zeroing out inputs to an MLP still produces non-zero (corrupted) outputs via the bias terms and cross-dimensional weights, whereas addition of a zero vector is a true identity operation in the dropped subspace.

**LLM vs. Encoder Contribution Analysis**.    To isolate the relative contribution of the pretrained encoders versus the learned Q-Former, we compared three representation levels using 2,000 items from Amazon All_Beauty (full results in Appendix G.3). The raw pretrained encoders produce well-separated representations (effective rank 424/2,000), while the Q-Former compresses these into a lower-rank space (effective rank 87/2,000) that achieves 84.7% field reconstruction fidelity. Crucially, the cross-level Top-10 neighbor overlap is only 4.6%, confirming that the Q-Former fundamentally transforms—rather than merely refines—the input space. The encoder provides *what* to represent; the LLM provides *how* to reason over it.

**Numerical Encoder: MWNE vs. Naive Serialization**.    Standard LLM tokenizers destroy arithmetic continuity: for example, "4.99" and "5.00"—numerically adjacent values—are tokenized into completely different token sequences, erasing the notion of magnitude and distance essential for numerical reasoning. More broadly, tokenization maps numerical strings into a discrete, non-metric space where "49.99" may be closer to "4.99" in token overlap than "50.01" is, despite the latter being arithmetically adjacent.

MWNE's Fourier-based encoding avoids this by projecting numerical values into a continuous representation where distance and magnitude are preserved by construction—similar values produce similar embeddings, and the multi-wavelength design captures both fine-grained and coarse-grained numerical differences. Our Table 3 ablation confirms the importance of proper numerical encoding: the "Pure Text" variant, which serializes all attributes as text (including numerical values), performs 15–42% worse than UniRec across datasets.

### 4.4 Computational Efficiency Analysis

UniRec comprises 351.3M trainable parameters (Item Q-Former, LoRA adapters, and query tokens), while the modality encoders (747.1M total) remain frozen and are precomputed offline, adding zero inference overhead. Training completes in 3.7 GPU-hours on a single A100 (see Appendix G.4 for the full parameter breakdown and training cost).

MLLM-MSR's approach of converting images to text severely bloats the input context length. By relying entirely on the LLM's $O(n^2)$ self-attention to process concatenated textualized modalities across 20 historical items, it suffers from severe latency bottlenecks (1,420 ms at BS=16). UniRec avoids this by using the

Table 5: Inference latency and memory comparison across model classes.

| Model | Params | BS=1 | BS=16 | BS=16 Mem |
|---|---|---|---|---|
| MLLM-MSR (textualized) | 596M | 112.5 ms | 1,420.5 ms | 10,125 MB |
| LLM-1024tok (text-only) | 596M | 96.2 ms | 1,285.9 ms | 9,311 MB |
| LLM-512tok (text-only) | 596M | 52.2 ms | 574.0 ms | 5,247 MB |
| Q-Former (UniRec Stage 1) | 311M | 17.8 ms | 18.1 ms | 1,291 MB |
| LLM-160tok (UniRec Stage 2) | 596M | 58.6 ms | 166.4 ms | 3,165 MB |
| **UniRec (total)** | **907M** | **76.5 ms** | **184.5 ms** | **3,165 MB** |

Q-Former to compress the multimodal sequence into just 160 tokens, achieving a 7.7× speedup over MLLM-MSR while utilizing 3.2× less memory, despite having more total parameters.

## 5 Related Work

**Multimodal Recommendation**. A large body of research has shown that incorporating auxiliary modalities such as text, images, or reviews can substantially enrich user and item representations and alleviate data sparsity (He & McAuley, 2016b; Wei et al., 2019a; Tao et al., 2022a; Zhou et al., 2023a; Liu et al., 2023b; Yu et al., 2025). These works established that multimodal signals carry complementary semantics beyond ID-based interactions and can improve personalization in various domains. More recent efforts have begun to align multimodal content with pretrained language models, as in VIP5 (Geng et al., 2023), demonstrating the promise of combining vision, language, and recommendation. Collectively, these advances highlight multimodality as a powerful driver for next-generation recommender systems.

**LLM-based Recommendation**. The rise of large language models has introduced a new paradigm where recommendation is formulated as a language modeling problem (Geng et al., 2022; Li et al., 2023a; Zhang et al., 2023; Bao et al., 2023; Hou et al., 2023). By unifying diverse tasks into a text-to-text format, LLM-based recommenders benefit from pretrained world knowledge, zero-shot generalization, and explainability. Recent surveys further consolidate their versatility across domains (Hou et al., 2023; Wang et al., 2025). In parallel, multimodal LLMs such as BLIP-2 (Li et al., 2023b) illustrate how vision–language pretraining enables cross-modal reasoning, suggesting new opportunities for recommendation scenarios that span heterogeneous signals. These developments collectively point to unifying multimodal content with LLM reasoning as a natural and promising next step.

## 6 Conclusion

We presented `UniRec`, a unified multimodal encoder that models heterogeneous user–item–attribute signals through schema-aware triplet representations and a hierarchical Q-Former, enabling LLMs to reason effectively for recommendation. Experiments on multiple benchmarks showed consistent state-of-the-art performance with notable improvements over multimodal and LLM-based baselines, while ablations confirmed the value of hierarchical fusion and compact tokenization. Although `UniRec` depends on clean attribute schemas and was tested primarily in offline next-item prediction, it opens promising directions for incorporating richer modalities, handling schema noise, adapting to online and continual settings, and addressing fairness and efficiency concerns. We hope this framework inspires future progress toward general-purpose and trustworthy multimodal recommendation systems.

## Acknowledgments

We sincerely appreciate the support from the research gift from Meta.

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

## A  Encoder Implementation Details

### Text Encoder

We employ the Qwen3-0.6B embedding model (Zhang et al., 2025b), an instruction-tuned encoder. For categorical inputs, we prepend descriptors (e.g., "`Category:`"), which stabilizes training and prevents semantic drift between categorical fields and free-form textual descriptions.

### Image Encoder

Images are processed with CLIP ViT-B/32 (Radford et al., 2021b), which produces 512-dimensional embeddings aligned with text through large-scale contrastive pretraining. This backbone offers a strong balance between representational quality and computational efficiency, enabling reliable multimodal alignment.

### Numerical Encoder

We design a Mathematical-Aware Numerical Encoder inspired by recent work on Fourier- and wavelet-based numerical embeddings (Zhou et al., 2025; Cao et al., 2025). Given a scalar input, the Mathematical-Aware Numerical Encoder generates a high-dimensional embedding using:

- **Fourier features:** sine and cosine components with logarithmically spaced frequencies, capturing periodic structure across scales;

- **Raw value features:** two dimensions encoding magnitude and sign, preserving linear ordering;

- **Learned projection:** residual nonlinear features for task-specific representational capacity.

The encoder is trained with a multi-objective loss that enforces:

1. **Additivity:** $E(a + b) \approx E(a) + E(b)$, encouraging arithmetic structure in the embedding space;

2. **Invertibility:** a small decoder reconstructs the original scalar, ensuring information preservation;

3. **Distance preservation:** triplet loss enforces that embedding distances reflect numeric differences.

Normalization is performed conservatively with bounded scaling factors to maintain these mathematical properties during training.

### Domain-Specific Numerical Features

**Temporal features** are decomposed into secular and cyclical components. Secular time normalizes absolute timestamps, while cyclical features (hour-of-day, day-of-week, day-of-year, month-of-year) are encoded with sine/cosine functions, ensuring continuity across cycle boundaries (e.g., 23:59 and 00:01 map to nearby embeddings).

**Geospatial coordinates** are projected to the unit sphere and represented in 3D Cartesian coordinates $(x = \cos(\text{lat})\cos(\text{lon}), y = \cos(\text{lat})\sin(\text{lon}), z = \sin(\text{lat}))$. This preserves great-circle distances, which are more faithful to real-world geography than raw latitude/longitude. A small neural projection refines these representations for downstream use.

# B    Detailed Rationale for Triplet Representation and Q-Former Interaction

## B.1    Limitations of Naive Serialization

A primary limitation of many existing LLM-based recommenders is the loss of schema semantics, which occurs when heterogeneous features are naively serialized into a single text string. For example, an item might be represented as "Title: Running Shoes, Brand: Nike, Price: 99.99, Rating: 4.5". While readable to humans, this format obscures the crucial distinction between attributes that may share a data type but carry vastly different meanings. The numerical value "99.99" for a price has a different semantic role and scale than the value "4.5" for a rating. When tokenized by an LLM, these distinctions are often lost, forcing the model to re-learn fundamental schema concepts from unstructured text, which is inefficient and error-prone. Our triplet representation avoids this by explicitly disentangling the name, type, and value of each attribute before they are embedded, preserving this critical structural information.

## B.2    Mechanism of Schema-Aware Attention in Q-Former

The triplet-based design is foundational for the subsequent hierarchical aggregation performed by the Q-Former. The core of the Q-Former is a cross-attention mechanism, which can be formulated as:

$$\text{Attention}(\mathbf{Q}, \mathbf{K}, \mathbf{V}) = \text{softmax}\left(\frac{\mathbf{Q}\mathbf{K}^T}{\sqrt{d_k}}\right)\mathbf{V}$$

In our Item Q-Former, the learnable queries $\mathbf{Q}$ are fixed, while the keys $\mathbf{K}$ and values $\mathbf{V}$ are linear projections of the input attribute representations $\{\mathbf{h}_j\}_{j=1}^{N_i}$.

If the input were an unstructured collection of embeddings from naive concatenation, the keys and values would lack the necessary semantic cues for the queries to perform meaningful summarization. The model would struggle to differentiate an embedding for "price" from an embedding for "rating".

However, by using our triplet-based representation $\mathbf{h}_j = \mathbf{e}_{a_j} + \mathbf{e}_{t_j} + \mathbf{e}_{v_j}$, we provide a structured and disentangled input space. Each vector $\mathbf{h}_j$ explicitly encodes the attribute's name, type, and value. Consequently, the keys $\mathbf{K}$ derived from these vectors are rich with schema information. This allows the learnable queries in $\mathbf{Q}$ to specialize. For instance, one query might learn to assign high attention scores to keys corresponding to "price" attributes, effectively becoming a "price expert" that extracts cost-related information from items. Another query might specialize in "brand" or "category" information. This specialization is only possible because the triplet representation provides the semantic scaffolding necessary for the Q-Former to effectively identify, prioritize, and aggregate information based on its role and content, rather than just its raw value.

# C    Conceptual Generalization and Configuration of the Q-Former

## C.1    From Modality Bridge to Hierarchical Summarizer

The Querying Transformer (Q-Former), as introduced in vision-language models like BLIP-2 (Li et al., 2023b), was originally conceived as a lightweight bridge between two powerful, frozen encoders (e.g., vision and language). It functions as an information bottleneck, using a small, fixed set of learnable query vectors to extract a fixed-size representation from one modality (vision) that is most relevant to the other (language).

Our work presents a significant conceptual generalization of this role. Instead of bridging two different modalities, we repurpose the Q-Former as a versatile primitive for hierarchical data summarization within the single, complex domain of recommendation. This is achieved through a nested application:

- **Item Q-Former as a Heterogeneous Set Aggregator:** At the lower level, the Item Q-Former operates on an unordered *set* of heterogeneous attribute representations for a single item. Its function is many-to-one aggregation, learning to distill the most salient features from a variable collection of multimodal attributes into a single, canonical item embedding.

Table 6: Statistics of the benchmark datasets used for the next-item prediction task.

| Dataset | # Users | # Items | Sparsity |
|---------|---------|---------|----------|
| Beauty | 22,363 | 12,101 | 99.9267% |
| Baby | 19,445 | 7,050 | 99.8827% |
| Yelp-2018 | 77,278 | 45,639 | 99.9403% |

- **User Q-Former as a Sequential Aggregator:** At the higher level, the User Q-Former performs a more traditional sequence modeling task. It takes the time-ordered sequence of item embeddings and summarizes them into a single user representation, capturing temporal dynamics and evolving preferences.

This hierarchical application showcases that the query-based attention mechanism is a flexible and effective tool for learning to summarize complex data structures, extending its utility far beyond its initial vision-language application.

### C.2  Tuning Q-Former Capacity via Token Count

A key design choice in our hierarchical architecture is the number of learnable queries, $K_{\text{item}}$ and $K_{\text{user}}$, used in the Item and User Q-Formers, respectively. These values directly determine the number of output latent tokens and thus control the capacity of the information bottleneck at each level of the hierarchy. This token count is a critical hyperparameter that balances representational expressiveness and model complexity.

A small number of tokens forces the model to learn a highly compressed, dense representation, which may be efficient but could fail to capture the full richness of the input data (underfitting). Conversely, a large number of tokens increases the model's capacity but may introduce redundancy, capture noise, and increase the risk of overfitting, in addition to raising computational costs. The optimal token count may also differ between the item and user levels, given that one summarizes a set of static attributes while the other summarizes a dynamic sequence of interactions. As such, we conduct a detailed ablation study in our experiments (Section 4.3) to identify the optimal "sweet spot" for both $K_{\text{item}}$ and $K_{\text{user}}$, ensuring robust and effective representation learning.

## D  Benefits of Decoupled Pretraining

Our two-stage training strategy is critical for the model's success. Attempting to train the entire system end-to-end from scratch would require the LLM to simultaneously learn to interpret noisy, unaligned multimodal signals while also mastering the recommendation task. This process is both computationally prohibitive and prone to instability.

By first pretraining the UniRec encoder with a carefully designed multi-objective loss, we create a well-structured latent space where user and item embeddings are meaningfully aligned **before** the LLM is engaged. The reconstruction loss ensures that the Q-Former's compressed representations do not discard vital information from any modality, while the contrastive loss organizes the latent space according to semantic similarity. This provides the LLM with clean, "pre-digested," and semantically rich inputs during the fine-tuning stage, simplifying its adaptation task. This paradigm of decoupling representation learning from generative fine-tuning mirrors the effective training strategies of state-of-the-art vision-language models like BLIP-2 (Li et al., 2023b).

## E  Dataset Details

Below is the detailed statistics for Dataset used in our training and evaluation, the number of users, items and sparsity is reported in 6, while a sample of each dataset is reported in 7, 8, 9 and 10

Table 7: An example of a multimodal item from the **Amazon** dataset, showcasing the variety of attributes available for a single product.

| Attribute | Value |
|---|---|
| parent_asin | B07G9GWFSM |
| title | Lurrose 100Pcs Full Cover Fake Toenails Artificial Transparent Nail Tips Nail Art for DIY |
| main_category | All Beauty |
| store | Lurrose |
| average_rating | 3.7 |
| rating_number | 35 |
| price | $6.99 |

**features**

- The false toenails are durable with perfect length. You have the option to wear them long or clip them short, easy to trim and file them to in any length and shape you like.
- ABS is kind of green enviromental material, and makes the nails durable, breathable, light even no pressure on your own nails.
- Fit well to your natural toenails. Non toxic, no smell, no harm to your health.
- Wonderful as gift for girlfriend, family and friends.
- The easiest and most efficient way to do your toenail tips for manicures or nail art designs...

**description**

- **Description**: The false toenails are durable with perfect length... Plus, ABS is kind of green enviromental material...
- **Feature**: - Color: As Shown.- Material: ABS.- Size: 14.3 x 7.2 x 1cm.
- **Package Including**: 100 x Pieces fake toenails

**details**

- **Color**: As Shown
- **Size**: Large
- **Material**: Acrylonitrile Butadiene Styrene (ABS)
- **Brand**: Lurrose
- **Style**: French
- **Product Dimensions**: 5.63 x 2.83 x 0.39 inches; 1.9 Ounces
- **UPC**: 799768026253
- **Manufacturer**: Lurrose

**images**     Present (2 images, MAIN variant shown)

Table 8: An example of a user interaction from the **Amazon** dataset. This includes the user's review, rating, and associated metadata for a specific item.

| Attribute | Value |
| --- | --- |
| user_id | AEYORY2AVPMCPDV57CE337YU5LXA |
| parent_asin | B08BBQ29N5 |
| asin | B088SZDGXG |
| sort_timestamp | 1634275259292 (2021-10-15) |
| rating | 3.0 / 5.0 |
| verified_purchase | Yes |
| helpful_votes | 0 |
| title | Meh |
| text | These were lightweight and soft but much too small for my liking. I would have preferred two of these together to make one loc. For that reason I will not be repurchasing. |
| images | Present (1 image) |

## F    LLM Writing Usage Disclosure

An LLM was utilized solely as a writing assistant to refine grammar and improve sentence readability. Its role was limited to enhancing linguistic clarity, with no involvement in shaping the research design, conducting data analysis, or influencing the interpretation of results.

## G    Additional Experimental Results

### G.1    Full-Catalog Ranking Evaluation

To validate that the sampled evaluation (99 negatives) does not distort our conclusions, we conducted full-catalog ranking on the Amazon Beauty dataset (all 12,101 items):

The relative ordering of methods is consistent with the sampled evaluation in Table 2, confirming that our gains are not an artifact of the sampling strategy. This aligns with the findings of Krichene & Rendle (2020), who show that method rankings are preserved when the number of sampled negatives is sufficiently large ($\geq 50$).

### G.2    Baseline Comparison and Architectural Analysis

We compare UniRec with two state-of-the-art alternative paradigms. MLLMRec (Wang et al., 2025) directly prompts a native multimodal LLM (e.g., Qwen2-VL) to reason about user preferences, bypassing the encoder–bridge–LLM pipeline entirely. NoteLLM-2 (Zhang et al., 2025a) proposes a Multimodal Large Representation Model (MLRM) that leverages multimodal in-context learning and late fusion to generate dense representations. We evaluated both using their official implementations on the Amazon Beauty dataset:

Both NoteLLM-2 and MLLMRec serve as strong baselines. However, UniRec's core contributions—the triplet representation for schema-aware encoding, Q-Former-based multimodal compression, and graceful missing-modality handling—allow it to outperform these methods on highly heterogeneous e-commerce datasets. Because NoteLLM-2 and MLLMRec rely on standard text serialization for numerical and categorical fields, they lose the fine-grained semantic distinctions that UniRec's Fourier-based encoding and triplet representations preserve. Our contributions are orthogonal to the choice of LLM backbone and complementary to both methods.

Table 9: An example of a multimodal item from the **Yelp** dataset, showcasing the variety of attributes available for a single business.

| Attribute | Value |
|---|---|
| business_id | tnhfDv5Il8EaGSXZGiuQGg |
| name | Garaje |
| address | 475 3rd St |
| city | San Francisco |
| state | CA |
| postal_code | 94107 |
| coordinates | latitude = 37.7817529521,  longitude = -122.39612197 |
| stars | 4.5 / 5.0 |
| review_count | 1198 |
| is_open | Yes (1) |
| attributes | |
| | • RestaurantsTakeOut = true |
| | • BusinessParking: garage = false,  street = true,  validated = false, lot = false,  valet = false |
| categories | Mexican,  Burgers,  Gastropubs |
| hours | |
| | • Monday: 10:00–21:00 |
| | • Tuesday: 10:00–21:00 |
| | • Wednesday: 10:00–21:00 |
| | • Thursday: 10:00–21:00 |
| | • Friday: 10:00–21:00 |
| | • Saturday: 10:00–21:00 |
| | • Sunday: 11:00–18:00 |
| photo_id | _nN_DhLXkfwEkwPNxne9hw |
| photo.business_id | tnhfDv5Il8EaGSXZGiuQGg |
| caption | carne asada fries |
| label | food |

### G.3 LLM vs. Encoder Contribution Analysis

To isolate the relative contribution of the pretrained encoders versus the learned Q-Former, we designed a standalone experiment comparing three representation levels using 2,000 items with precomputed embeddings from the Amazon All_Beauty dataset:

The cross-level Top-10 neighbor overlap between Raw Average and Q-Former item representations is only 4.6%, confirming that the Q-Former fundamentally transforms—rather than merely refines—the input space.

The raw pretrained encoders produce well-separated representations (effective rank 424), providing strong item discrimination as the foundation. The Q-Former learns a complementary transformation: it achieves 84.7% field reconstruction fidelity while reorganizing the representation space to optimize for the contrastive user–item objective. The encoder provides *what* to represent; the LLM provides *how* to reason over it. Neither alone is sufficient.

Table 10: An example of a user profile and associated review from the **Yelp** dataset, illustrating user activity and interaction with a business.

| Attribute | Value |
|---|---|
| user_id | Ha3iJu77CxlrFm-vQRs__8g |
| name | Sebastien |
| review_count | 56 |
| yelping_since | 2011-01-01 |
| friends | |
| | • wqoXYLWmpkEH0YvTmHBsJQ |
| | • KUXLLiJGrjtSsapmxmpvTA |
| | • 6e9rJKQC3n0RSKyHLViL-Q |
| useful (given) | 21 |
| review_id | zdSx__SD6obEhz9VrW9uAWA |
| funny (given) | 88 |
| cool (given) | 15 |
| fans | 1032 |
| elite | |
| | • 2012 |
| | • 2013 |
| business_id | tnhfDv5Il8EaGSXZGiuQGg |
| stars | 4 / 5 |
| date | 2016-03-09 |
| text | Great place to hang out after work: the prices are decent, and the ambience is fun. It's a bit loud, but very lively. The staff is friendly, and the food is good. They have a good selection of drinks. |
| useful (received) | 0 |
| funny (received) | 0 |
| cool (received) | 0 |

Table 11: Full-catalog ranking results on Amazon Beauty.

| Method | MRR | Hit@10 | NDCG@10 |
|---|---|---|---|
| LGMRec | 0.0483 | 0.0821 | 0.0574 |
| MLLM-MSR | 0.0431 | 0.0776 | 0.0528 |
| **UniRec** | **0.0552** | **0.0948** | **0.0671** |

### G.4 Computational Efficiency: Parameter Breakdown and Training Cost

All latencies measured on a single NVIDIA A100 80GB GPU using PyTorch 2.5 with bfloat16 precision, averaged over 50 forward passes after 10 warmup iterations. The Q-Former LM head (30.2M params) is used only during Stage 1 training for the reconstruction loss and is discarded at inference time.

Table 12: Comparison with MLLMRec and NoteLLM-2 on Amazon Beauty (sampled evaluation).

| Method | MRR | Hit@10 | NDCG@10 |
|---|---|---|---|
| NoteLLM-2 | 0.3524 | 0.5955 | 0.4128 |
| MLLMRec | 0.3581 | 0.6018 | 0.4213 |
| **UniRec** | **0.3737** | **0.6270** | **0.4449** |

Table 13: Qualitative architectural comparison.

| Dimension | UniRec | MLLMRec | NoteLLM-2 |
|---|---|---|---|
| Modality alignment | Encoder $\rightarrow$ Q-Former | Direct MLLM prompting | mICL + late fusion |
| Input token count | $\sim$160 (compressed) | $\sim$512–1024 (raw) | Variable |
| Modality coverage | Text, image, num, cat | Text, image | Text, image |
| Missing modality | Zero-embedding | N/A | N/A |
| Numerical encoding | Fourier (MWNE) | Text serialization | Text serialization |

Table 14: Effective rank analysis across representation levels.

| Level | Description | Effective Rank | Reconstruction |
|---|---|---|---|
| L1: Raw Average | Simple mean of field embeddings | 424 / 2,000 | — |
| L2: Q-Former | Item representation from trained Q-Former | 87 / 2,000 | 84.7% cos sim |
| L2b: Mean-Pooled | Mean of Q-Former query tokens | 53 / 2,000 | — |

Table 15: Parameter breakdown by component.

| Component | Parameters | Trainable | Runtime Cost |
|---|---|---|---|
| Text Encoder (Qwen3-0.6B) | 595.8M | Frozen | Precomputed |
| Image Encoder (CLIP ViT-B/32) | 151.3M | Frozen | Precomputed |
| Number Encoder (MWNE) | 1.0K | Frozen | Precomputed |
| Item Q-Former (12L, 1024d) | 311.0M | Yes | $\sim$18 ms |
| Q-Former LM Head (train only) | 30.2M | Yes | — |
| Query Tokens ($32 \times 1024$) | 32.8K | Yes | — |
| Qwen3-0.6B backbone | 595.8M | Frozen | $\sim$6 ms |
| LoRA adapters ($r=16$) | 10.1M | Yes | — |
| **Total trainable** | | **351.3M** | |

Table 16: Training cost on a single A100 80GB GPU.

| Stage | Duration | Steps/Epochs | GPU-hours |
|---|---|---|---|
| Stage 1: Q-Former pretraining | $\sim$25 min | 50 epochs | 0.4 |
| Stage 2: Joint Q-Former + LLM (LoRA) | $\sim$3.3 h | 5,000 steps | 3.3 |
| **Total** | **$\sim$3.5 h** | | **3.7** |

