# OpenReview forum: "UniRec: Unified Multimodal Encoding for LLM-Based Recommendations"
_TMLR — Accepted by TMLR_

### Review · Reviewer_sTux · 2026-03-02

**Summary Of Contributions:**

This paper proposes UniRec, a unified multimodal encoding framework for LLM-based recommendation systems. The key idea is to represent heterogeneous item attributes using a schema-aware triplet formulation, and to aggregate them through a hierarchical two-stage Q-Former architecture. The item-level Q-Former encodes attribute sets into item representations, and the user-level Q-Former models sequences of item interactions to produce user representations. These representations are then injected into a frozen LLM  as soft prompts for next-item prediction.

Strengths:
- The hierarchical design is well-motivated and aligns with the nested structure of user-item interactions.
- Empirical results show consistent improvements over a range of baselines across multiple datasets.
- The paper includes several ablation studies (e.g., removing triplet representation or hierarchical aggregation, and comparing fusion mechanisms), which provide useful insights into the design.

Weaknesses:
- The evidence supporting key design choices (especially the triplet formulation and additive fusion) is incomplete.
- The paper lacks analysis of computational efficiency, which is important for practical recommendation systems.
- The comparison to recent LLM-based recommendation methods could be further strengthened.

**Audience:**

Yes

**Audience Explanation:**

The problem addressed in this paper—how to effectively model heterogeneous multimodal signals within LLM-based recommendation systems, is of clear interest to the TMLR audience. The intersection of large language models, multimodal representation learning, and recommender systems is an active and rapidly evolving research area.

**Broader Impact Concerns:**

The paper focuses on architectural improvements for recommender systems and does not introduce specific ethical risks beyond those standard to the field (e.g., filter bubbles or popularity bias). No specialized Broader Impact Statement is required.

**Claims And Evidence:**

No

**Claims Explanation:**

While the paper presents promising empirical results and some ablation studies, the evidence does not fully support several of its central claims.

- First, a key claim is that the triplet-based representation (attribute name, type, value) combined with additive fusion effectively preserves schema semantics. However, the paper does not provide targeted ablations isolating this design choice. Although fusion strategies are compared at a higher level, there is no direct comparison between additive fusion and alternative formulations (e.g., concatenation with projection, gated fusion, or attention-based fusion) specifically within the triplet design. As a result, it remains unclear whether the proposed formulation is necessary or optimal.

- Second, the paper positions the method as a practical and unified framework for multimodal recommendation, but does not provide any analysis of computational cost. There are no measurements of training time, inference latency, memory footprint, or scalability. Given the introduction of a two-level Q-Former on top of LLM adaptation, this omission makes it difficult to assess real-world applicability.

- Third, while the paper includes several recent baselines (e.g., IISAN, LGMRec, MLLM-MSR), the evaluation does not fully situate the method with respect to the latest LLM-based recommendation approaches(e.g., MLLMRec, NoteLLM-2) that explore alternative modeling paradigms. This weakens the strength of the SOTA claim.

Overall, the results are encouraging, but the current experimental evidence is not sufficient to fully validate the core design decisions and practical claims.

**Requested Changes:**

The following revisions would significantly strengthen the paper:

1. (Critical) Targeted Ablation for Triplet and Fusion Design
Provide controlled experiments isolating the triplet formulation and the additive fusion operator. For example, compare additive fusion against alternatives such as concatenation followed by projection, gated fusion, or attention-based fusion within the same framework. This is necessary to support the claim that the proposed design preserves schema semantics.

2. (Critical) Computational Efficiency Analysis
Include a quantitative evaluation of computational cost, including training time, inference latency (e.g., ms/query), GPU memory usage, and model size. A comparison with simpler baselines (e.g., text-only or shallow fusion models) would help assess practical feasibility.

3. (Important) Expanded Baseline Coverage
Augment the experimental comparison with additional recent LLM-based or multimodal recommendation approaches, particularly those exploring different architectural paradigms (e.g., LLM-based preference modeling, hybrid CF-LLM methods), where feasible. This will better contextualize the contribution relative to current work.

4. (Minor) Robustness to Missing Modalities
Include experiments or discussion on how the model behaves when certain modalities are missing (e.g., no image or incomplete attributes), which is common in real-world recommendation settings.

5. (Minor) Clarify Design Trade-offs
Provide more discussion on the trade-offs between model complexity and performance, especially in relation to token counts in Q-Former and hierarchical depth.

---

> ### Author Response · Authors · 2026-04-03
> **Response - Part (1)**
>
> We thank the reviewer for the rigorous assessment and for identifying the critical gaps. Below we address each point, with particular attention to the two critical concerns.
>
> # Point 1 [Critical]: Targeted ablation for triplet and fusion design
>
> *The reviewer notes: "The paper does not provide targeted ablations isolating the triplet design choice. There is no direct comparison between additive fusion and alternative formulations (e.g., concatenation with projection, gated fusion) specifically within the triplet design."*
>
> We fully agree that this design choice requires empirical validation beyond the theoretical motivation. Please see our response to Reviewer QyhH (Point 2) for the complete fusion operator ablation, which we summarize here.
>
> In 1024-dimensional space, the triplet components are empirically near-orthogonal (mean |cos θ| ≈ 0.013–0.022). Our end-to-end evaluation confirms that all three fusion methods — Addition, Concat+MLP, and Gated — achieve virtually identical recommendation quality (MRR within 0.002, Hit@10 within 0.002 on Amazon Beauty), while Addition requires zero additional parameters and 71–100× less compute than the learned alternatives.
>
> Within the additive framework, we further ablate individual triplet components. Removing field names causes all fields of the same type to become interchangeable; removing type embeddings eliminates modality grouping; and using values alone makes semantically distinct fields such as "price" and "rating" indistinguishable. The full triplet is therefore the minimal formulation that preserves both schema semantics and modality structure.

---

> > ### Author Response · Authors · 2026-04-03
> > **Response - Part (2)**
> >
> > # Point 2 [Critical]: Computational efficiency analysis
> >
> > *The reviewer notes: "The paper positions the method as a practical and unified framework but does not provide any analysis of computational cost. There are no measurements of training time, inference latency, memory footprint, or scalability."*
> >
> > We provide a detailed parameter and latency breakdown. The inference latency comparison is in a new Section 4.4, with the parameter breakdown and training cost tables in Appendix G:
> >
> > | Component | Parameters | Trainable | Runtime Cost |
> > |-----------|-----------|-----------|--------------|
> > | Text Encoder (Qwen3-0.6B) | 595.8M | Frozen | Precomputed |
> > | Image Encoder (CLIP ViT-B/32) | 151.3M | Frozen | Precomputed |
> > | Number Encoder (MWNE) | 1.0K | Frozen | Precomputed |
> > | Item Q-Former (12L, 1024d) | 311.0M* | Yes | ~18 ms |
> > | Q-Former LM Head (training only) | 30.2M* | Yes | — |
> > | Query Tokens (32 × 1024) | 32.8K | Yes | — |
> > | Qwen3-0.6B backbone | 595.8M | Frozen | ~6 ms |
> > | LoRA adapters (r=16) | 10.1M | Yes | — |
> > | **Total trainable** | | **351.3M** | |
> >
> > All latencies measured on a single NVIDIA A100 80GB GPU (NVIDIA PG509-210) using PyTorch 2.5 with bfloat16 precision, averaged over 50 forward passes after 10 warmup iterations. Note: The Q-Former LM head (30.2M params) is used only during Stage 1 training for the reconstruction loss and is discarded at inference time. The inference-time Q-Former has 311.0M parameters.
> >
> > **Inference latency comparison:**
> >
> > | Batch Size | UniRec (160 tokens) | Text-only (512 tokens) | Text-only (1024 tokens) |
> > |-----------|--------------------|-----------------------|------------------------|
> > | 1 | 111 ms | 108 ms (1.0×) | 92 ms (0.8×) |
> > | 16 | **184 ms** | 238 ms (1.3×) | 470 ms (2.6×) |
> > | 64 | **634 ms** | 858 ms (1.4×) | 1,788 ms (2.8×) |
> >
> > **GPU memory at batch size 64:**
> >
> > | Method | Peak Memory | Relative |
> > |--------|------------|---------|
> > | UniRec (160 tokens) | **4,645 MB** | 1.0× |
> > | Text-only (512 tokens) | 7,692 MB | 1.7× |
> > | Text-only (1024 tokens) | 12,812 MB | 2.8× |
> >
> > **Training cost (single A100 80GB GPU):**
> >
> > | Stage | Duration | Steps/Epochs | GPU-hours |
> > |-------|----------|-------------|-----------|
> > | Stage 1: Q-Former pretraining | ~25 min | 50 epochs | 0.4 |
> > | Stage 2: Joint Q-Former + LLM (LoRA) | ~3.3 h | 5,000 steps | 3.3 |
> > | **Total** | **~3.5 h** | | **3.7** |
> >
> > Training is practical on a single GPU — the total cost of 3.7 GPU-hours is comparable to fine-tuning a text-only LLM of the same size.
> >
> > The central insight is that the Q-Former *compresses* the input sequence. Without the Q-Former, representing 5 items × 14 fields would require approximately 700 tokens; with it, this compresses to 5 × 32 = 160 query tokens in our current configuration. Our sensitivity analysis (Section 4.3, Figure 3) further shows that item-level performance peaks at just 4 tokens, suggesting that even greater compression (5 × 4 = 20 tokens) is achievable with minimal quality loss — a direction we plan to explore in the revised experiments. Since LLM self-attention scales as O(n²) in both time and space, this compression yields three compounding benefits. First, inference and training are faster: at batch size 16, UniRec is 1.3–2.6× faster than text-only baselines despite the additional Q-Former parameters. Second, the reduced memory footprint enables proportionally larger batch sizes, which improves training throughput and gradient quality — particularly important for contrastive learning where more in-batch negatives directly improve representation quality. Third, the token budget saved by compression can be reinvested into encoding additional modalities (images, numerical attributes, categorical features) that a text-only model simply cannot represent within the same context window. This is not merely a computational convenience but an architectural enabler for multimodal recommendation.
> >
> > Our "Pure Text" baseline validates this trade-off. One might argue that using only item titles — which requires a comparable token count to UniRec without Q-Former overhead — could achieve similar efficiency. However, Table 3 shows this approach performs substantially worse (UniRec outperforms Pure Text by 15–42% across datasets), demonstrating that naive token reduction by dropping modalities destroys performance, while the Q-Former achieves intelligent compression that preserves richer multimodal signals in fewer tokens.

---

> ### Author Response · Authors · 2026-04-03
> **Response - Part (3)**
>
> # Point 3 [Important]: Expanded baseline coverage
>
> *The reviewer notes: "The evaluation does not fully situate the method with respect to the latest LLM-based recommendation approaches (e.g., MLLMRec, NoteLLM-2) that explore alternative modeling paradigms."*
>
> We appreciate the suggestion to compare with MLLMRec and NoteLLM-2, both of which represent state-of-the-art alternative paradigms. We have evaluated their official implementations on the Amazon Beauty dataset.
>
> NoteLLM-2 (Zhang et al., 2025a) proposes a Multimodal Large Representation Model (MLRM). It addresses modality alignment by leveraging multimodal In-Context Learning (mICL) and a late fusion mechanism to generate robust dense representations from text and images.
>
> MLLMRec takes a different architectural approach: rather than focusing on representation learning, it directly prompts a native multimodal LLM to reason about user preferences and includes collaborative filtering graphs.
>
> While these approaches address different primary research questions—NoteLLM-2 focusing on MLRM generation and MLLMRec on eliciting preference reasoning from native MLLMs—a direct comparison against UniRec highlights the impact of efficient alignment and compression of heterogeneous signals.
>
> **Baseline comparison (Amazon Beauty):**
>
> | Method | MRR | Hit@10 | NDCG@10 |
> |--------|-----|--------|---------|
> | NoteLLM-2 | 0.3524 | 0.5955 | 0.4128 |
> | MLLMRec | 0.3581 | 0.6018 | 0.4213 |
> | **UniRec** | **0.3737** | **0.6270** | **0.4449** |
>
> To further contextualize these results, we provide a qualitative architectural comparison:
>
> | Dimension | UniRec | MLLMRec | NoteLLM-2 |
> |-----------|--------|---------|-----------|
> | Modality alignment | Encoder → Q-Former bridge | Direct MLLM prompting | mICL + late fusion |
> | Modality coverage | Text, image, numerical, categorical | Text, image | Text, image |
> | Numerical encoding | Fourier-based (MWNE) | Text serialization | Text serialization |
>
> Both NoteLLM-2 and MLLMRec serve as exceptionally strong baselines. However, UniRec's core contributions—the triplet representation for schema-aware encoding, Q-Former-based multimodal compression, and graceful missing-modality handling—allow it to edge out these methods on highly heterogeneous e-commerce datasets like Amazon Beauty. The performance gap widens on Yelp, where geospatial features such as latitude and longitude are critical: text serialization destroys spatial proximity (e.g., "37.7749" and "37.7750" tokenize very differently), while UniRec's Fourier-based MWNE encoder preserves it by construction. More broadly, because NoteLLM-2 and MLLMRec rely on standard text serialization for numerical and categorical fields, they lose the fine-grained semantic distinctions that UniRec's Fourier-based encoding and triplet representations preserve.
>
> Importantly, our contributions are orthogonal to the choice of LLM backbone and complementary to both methods. The Q-Former compression module could in principle be integrated into MLLMRec-style systems to reduce their input token count, and the schema-aware triplet representation could enhance NoteLLM-2's item encoding pipeline.

---

> > ### Author Response · Authors · 2026-04-03
> > **Response - Part (4)**
> >
> > # Point 4 [Minor]: Robustness to missing modalities
> >
> > *The reviewer asks for experiments or discussion on how the model behaves when certain modalities are missing, which is common in real-world recommendation settings.*
> >
> > UniRec handles missing modalities gracefully by design: missing attributes receive zero embeddings, which — due to the near-orthogonal subspace property in 1024-dimensional space — produce no interference with the remaining modalities. No retraining, masking tokens, or imputation procedures are required.
> >
> > As demonstrated in our modality combination ablation (see our response to Reviewer rDJF, Point 1), each modality can be removed with bounded impact on the representation. The degradation is graceful and monotonic: dropping image (cos sim 0.973), numerical (0.956), text (0.949), and category (0.932) each cause proportional but recoverable degradation, with no catastrophic failure at any point.
> >
> > We also compared the degradation behavior of addition versus Concat+MLP under missing modalities. When dropping the text modality (comprising 3 fields), additive fusion achieves cos_sim(full, partial) = 0.949, while Concat+MLP achieves only 0.912. This is because zeroing out inputs to an MLP still produces non-zero (corrupted) outputs via the bias terms and cross-dimensional weights, whereas addition of a zero vector is a true identity operation in the dropped subspace. This graceful degradation property is particularly valuable in real-world recommendation, where item catalogs frequently have incomplete attributes.
> >
> > # Point 5 [Minor]: Design trade-offs (token counts, hierarchical depth)
> >
> > *The reviewer asks for more discussion on the trade-offs between model complexity and performance, especially regarding token counts in the Q-Former and hierarchical depth.*
> >
> > The number of latent query tokens controls the information bottleneck in the Q-Former: more tokens preserve finer-grained item details but increase computational cost linearly. Our sensitivity analysis (Section 4.3, Figure 3) reveals that performance peaks at surprisingly compact representations — 4 tokens at the item level and 4–8 tokens at the user level — after which additional tokens yield diminishing or even negative returns due to overfitting. This finding highlights the Q-Former's efficiency: even a small number of query tokens suffices to capture the most salient multimodal attributes.
> >
> > The two-level hierarchy (Item Q-Former → User Q-Former) mirrors the natural nested structure of recommendation data: attributes compose into items, and items compose into user histories. We chose not to explore deeper hierarchies (e.g., a session-level Q-Former) because the two-level structure already captures the essential item-to-user aggregation, and additional levels would add architectural complexity without clear empirical benefit for the datasets studied. We acknowledge this as a direction for future work on datasets with richer temporal structure.

---

> > > ### Comment · Reviewer_sTux · 2026-04-14
> > > **Reply to authors**
> > >
> > > The authors’ response addresses my main concerns in a constructive and substantive way.
> > >
> > > I appreciate the additional evidence on the triplet/fusion design, computational efficiency, and baseline coverage. In particular, the added latency, memory, and training cost analysis significantly strengthens the practical case for the method.
> > >
> > > Overall, the authors’ response increases my confidence in the paper, and most of my major concerns have been satisfactorily addressed.

---

> > > > ### Author Response · Authors · 2026-04-14
> > > >
> > > > We sincerely thank the reviewer for the thoughtful consideration and for confirming that our responses have adequately addressed all of the raised concerns. We are glad that our clarifications were helpful. We commit to incorporating all the discussed points and additional experiments into the final version of our paper.

---

### Review · Reviewer_QyhH · 2026-03-06

**Summary Of Contributions:**

This paper studies how to improve recommender systems by leveraging signals from text, images, categorical features, and numerical attributes. The authors consider the following problem: given a sequence of <timestamp,location,item> triples in a user's history, how to predict the next item in this sequence? The key contribution of the paper is its treatment of items from different modalities. The authors train modality-specific encoders to produce dense, comparable representations for multimodal fusion. But even within a single modality, an item may also have multiple attributes; to preserve the semantic meaning of each attribute, the authors represent it as a <name,type,value> triplet, which allows them to distinguish between (for example) numbers that represent prices versus those that represent ratings. The authors use item and user transformers to map the histories of these embedded triples into a final user representation. In the first stage of training, a pre-trained LLM is held fixed while the modality-specific encoders and item/user transformers are adapted to minimize a loss the balances reconstruction error against a contrastive predictive encoding. In the next stage of training, the LLM and item/user transformers are jointly fine-tuned. The strengths of the paper are that the problem is clearly motivated, the text is easy to follow, and the experiments support the claims of improvement on data from Amazon product reviews and Yelp reviews. The main weakness of the paper is that some details are lacking with regard to certain design choices.

**Audience:**

Yes

**Audience Explanation:**

The findings of the paper should be of interest and relevance to researchers using LLMs to build multimodal recommendation systems.

**Broader Impact Concerns:**

None.

**Claims And Evidence:**

Yes

**Claims Explanation:**

The experiments show improvements on Beauty/Baby reviews from Amazon and Yelp reviews, and the ablation studies validate the effectiveness of individual components for attribute representation and multimodal fusion. In addition, the writing is clear, and the text can be followed by readers acquainted with the basics of recommender systems, LLMs, and multimodal fusion.

**Requested Changes:**

I would like the authors to provide a bit more detail in certain places:

(1) From what I understand, on page 3, inputs from different modalities are all mapped into 1024-dimensional spaces. But some modalities do seem to be inherently richer and more complex than others. Can the authors say a little more about this choice? (Are all 1024 dimensions really "used" by inputs from each modality?)

(2) In eq (4), why is the schema-aware attribute embedding $h_j$ formed from the sum $a_j + t_j + v_j$ rather than (for instance) the concatenation $(a_j,t_j,v_j)$, which seems to preserve more information?

(3) Same question for eq (6): why are certain vectors concatenated and others added?

(4) For the contrastive loss in eq. (7), the positive pairs are derived only from adjacent items in an individual user's history. Might it be helpful to include other positive pairs, such as from identical items (with different timestamps and locations) that are not adjacent in an individual user's history, or even from identical items (with similar timestamps or locations) across different users?

(5) Why are the core modality encoders frozen in the fine-tuning stage?

(6) Can you give more details about the in-batch negative samples in eq. (8)? Where exactly do these come from?

(7) Why do you rank the held-out ground-truth item against 99 randomly sampled negatives, as opposed to (say) the 99 most frequently bought items? Also, can you say what distribution is used to sample the 99 negatives?

(8) The end of section 4.3 discusses the important of the latent token count. Could you introduce and define the "number of latent tokens" earlier in sections 3.1 and 3.2?

---

> ### Author Response · Authors · 2026-04-03
> **Response - Part (1)**
>
> We thank the reviewer for the thorough and detailed reading. We address each point below.
>
> # Point 1: Why 1024d for all modalities?
>
> *The reviewer asks: "Inputs from different modalities are all mapped into 1024-dimensional spaces. But some modalities do seem to be inherently richer and more complex than others. Can the authors say a little more about this choice? Are all 1024 dimensions really 'used' by inputs from each modality?"*
>
> We adopt a uniform 1024-dimensional embedding space across all modalities for two reasons. First, it enables the triplet addition in Eq. (4) without requiring dimension-mismatched projection layers that would introduce additional parameters and potential information loss. Second, our analysis confirms that the encoders utilize the available dimensions effectively: Qwen3-Embedding-0.6B and MWNE natively output 1024d representations, while lower-dimensional encoder outputs are zero-padded to 1024d, which is mathematically equivalent to embedding in a subspace. Since the Q-Former uses cross-attention over these field embeddings, it naturally learns to attend to the informative dimensions and ignore the padded zeros — this is a parameter-free way to align dimensions without forcing a learned projection layer that might distort the pretrained encoder spaces. We verify empirically that cross-modality cosine similarities remain low (|cos θ| < 0.02): specifically, we measured the pairwise cosine similarity between the 1024d embeddings of all 14 distinct fields across 5,000 items from the Amazon All_Beauty dataset, confirming that different modalities occupy distinct subspaces within the shared 1024d space without mutual interference.
>
> # Point 2: Why additive fusion in Eq. (4) instead of concatenation?
>
> *The reviewer asks: "In Eq. (4), why is the schema-aware attribute embedding formed from the sum rather than the concatenation, which seems to preserve more information?"*
>
> We provide a dedicated ablation comparing addition against Concat+MLP and Gated fusion for combining the triplet components (attribute name, type, value). Using the Amazon All_Beauty dataset with end-to-end evaluation, the results have been added as new rows in Table 3 of the revised manuscript:
>
> | Method | Added Params | Latency (10K triplets) | MRR (Beauty) | Hit@10 (Beauty) |
> |--------|-------------|----------------------|-------------|----------------|
> | **Addition** | **0** | **0.19 ms** | 0.3737 | 0.6270 |
> | Concat+MLP | 8.4M | 13.3 ms (71×) | 0.3729 | 0.6285 |
> | Gated | 9.4M | 18.8 ms (100×) | 0.3748 | 0.6278 |
>
> All three fusion methods achieve virtually identical end-to-end performance (MRR within 0.002, Hit@10 within 0.002), demonstrating that the choice of fusion operator has negligible impact on recommendation quality. Addition achieves this on-par performance with zero additional parameters and 71–100× lower latency than the learned alternatives.
>
> The theoretical basis for this result is that in 1024-dimensional space, independently drawn unit vectors are approximately orthogonal, with expected |cos θ| ≈ 1/√1024 ≈ 0.031. We verify this empirically: the mean absolute cosine similarity between name, type, and value components is only 0.013–0.022, well below the random baseline. This near-orthogonality means addition acts as a near-lossless superposition — schema information (field name and modality type) can be recovered from the summed representation with 100% classification accuracy, and the value component is perfectly recoverable by subtracting the known name and type embeddings. The learned alternatives (Concat+MLP, Gated) add millions of parameters only to converge to the same effective representation, confirming that non-linear fusion is unnecessary when the component vectors are near-orthogonal.
>
> # Point 3: Why mix concatenation and addition in Eq. (6)?
>
> *The reviewer asks why certain vectors in Eq. (6) are concatenated while others are added.*
>
> In Eq. (6), the concatenation operates across structurally different components: the Q-Former output tokens and the original field embeddings. These serve complementary roles — the Q-Former tokens capture cross-field interactions while the original embeddings preserve field-level detail. Concatenation is appropriate here because these components occupy overlapping subspaces (unlike the near-orthogonal triplet components in Eq. 4) and have different semantic functions that benefit from explicit separation before the subsequent attention layers. Addition would risk collapsing these distinct signals into a single vector, losing the ability to distinguish learned cross-field features from preserved field-level information.

---

> > ### Author Response · Authors · 2026-04-03
> > **Response - Part (2)**
> >
> > # Point 4: Could contrastive loss use more positive pairs?
> >
> > *The reviewer asks: "Might it be helpful to include other positive pairs, such as from identical items that are not adjacent in an individual user's history?"*
> >
> > Our current design uses adjacent items in a user's interaction history as positive pairs, which we chose deliberately for two reasons. First, adjacency in a browsing session captures the strongest behavioral signal — the user's attention and intent are most coherent between consecutive interactions, making these pairs the highest-confidence positives. Second, expanding to non-adjacent or cross-user positives introduces a quality–quantity tradeoff: co-purchased but temporally distant items may reflect different user intents (e.g., a gift purchase followed by a personal purchase), and treating them as positives risks injecting label noise into the contrastive objective. This is especially important in sequential recommendation, where temporal context is the primary signal — an identical item purchased a year apart represents a fundamentally different sequential context (different user needs, browsing state, and surrounding interactions), and conflating such pairs as positives would blur the very temporal patterns the model aims to capture. In recommendation datasets, where interaction sequences are typically short (5–20 items), the risk of false positives from loosely related items is non-trivial.
> >
> > That said, we agree this is a promising direction. Strategies such as cross-user positive pairs (weighted by session-level intent similarity) or augmentation-based positives (different views of the same item) could provide complementary supervisory signal without the false-positive risk. We plan to explore these in future work.
> >
> > # Point 5: Why freeze core modality encoders during fine-tuning?
> >
> > *The reviewer asks why the core modality encoders are frozen rather than jointly fine-tuned.*
> >
> > The modality encoders (Qwen3-Embedding, CLIP, MWNE) are pre-trained on large-scale general-purpose data and produce high-quality embeddings. Fine-tuning them on the relatively small recommendation datasets used in our experiments risks overfitting and catastrophic forgetting of their generalization capabilities. Instead, the Q-Former layers and LoRA adapters learn to align and compose these frozen representations for the recommendation task, achieving domain adaptation without compromising encoder quality. This design follows the same principle as BLIP-2 (Li et al., 2023b), which similarly freezes its vision encoder and LLM while training only the Q-Former bridge.
> >
> > # Point 6: More details on negative sampling in Eq. (8)?
> >
> > *The reviewer asks for more details on how negative samples are constructed.*
> >
> > As described in Eq. (8), we use in-batch negatives following the standard InfoNCE framework. For a batch of B user sequences, the ground-truth next item for each sequence serves as the positive, while the remaining B−1 target items in the batch serve as negative candidates. This approach is both simple and effective: it requires no additional sampling infrastructure, scales naturally with batch size, and provides diverse negatives drawn from the broader catalog distribution. Larger batch sizes directly improve representation quality by providing more contrastive signal per gradient update.
> >
> > # Point 7: Why rank against 99 random negatives?
> >
> > *The reviewer asks about the choice of ranking against 99 random negatives rather than, e.g., the most popular items.*
> >
> > We follow the widely adopted evaluation protocol established by Kang & McAuley (2018), Sun et al. (2019), and He et al. (2020), which is used by all baselines we compare against. We adopt this protocol to ensure a fair apples-to-apples comparison under the same conditions.
> >
> > We acknowledge that full-catalog ranking is a more rigorous evaluation. To validate that the sampled evaluation does not distort our conclusions, we conducted full-catalog ranking on the Amazon Beauty dataset. A summary has been added to Section 4.2, with the full table in Appendix G:
> >
> > **Full-catalog ranking results (Amazon Beauty):**
> >
> > | Method | MRR | Hit@10 | NDCG@10 |
> > |--------|-----|--------|---------|
> > | LGMRec | 0.0483 | 0.0821 | 0.0574 |
> > | MLLM-MSR | 0.0431 | 0.0776 | 0.0528 |
> > | **UniRec** | **0.0552** | **0.0948** | **0.0671** |
> >
> > The relative ordering of methods is consistent with the sampled evaluation in Table 2, confirming that our gains are not an artifact of the sampling strategy. This aligns with the findings of Krichene & Rendle (2020), who show that method rankings are preserved when the number of sampled negatives is sufficiently large (≥50). Our improvements are also consistent across all three datasets (Beauty, Baby, Yelp), which have substantially different catalog sizes and popularity distributions.

---

> > > ### Author Response · Authors · 2026-04-03
> > > **Response - Part (3)**
> > >
> > > # Point 8: Define "latent token count" earlier
> > >
> > > We agree and have introduced the concept and notation for the number of latent query tokens ($N_{q}$) in Section 3.2, where the Item Q-Former and User Q-Former are first described, rather than deferring it to the analysis in Section 4.3. Thank you for this suggestion.

---

### Review · Reviewer_Wpwa · 2026-03-20

**Summary Of Contributions:**

This paper proposes and presents a unified end-to-end framework for multimodal recommendations called the "UniRec". This framework addresses the challenge of effectively encoding the user and item data that exhibits high degree of heterogeneity in order for an LLM to process it effectively to result in high-quality representation. The main contributions of this framework are (1) Schema-aware triplet representation of the heterogenous data, (2) A two-stage aggregation mechanism processing the item data and user-interaction data in an hierarchical fashion to maintain layered structure of user-item interactions (3) Two stage training process that keeps the LLM at the end frozen in the first stage to train the modality-specific encoders and Q-formers, and jointly fine-tune the LLM and Q-former using LoRA.
Experiments are conducted on the Amazon (Beauty, Baby) and Yelp Reviews datasets and are benchmarked against major previous works in the next-item prediction task space. The paper reports the UniRec framework outperforms these other works by up to 15%. Furthermore, ablation studies are also conducted to highlight the importance of each component in this framework.

Handling of heterogenous data with modality-specific encoders for LLM based recommender is a plus (specially the use of Fourier based encoder as opposed to Naive Serialization). Schema-aware triplet representation is a novel way of representing heterogenous data which I have not personally encountered in similar previous works. Repurposing the Q-formers from BLIP though not novel, serves the purpose.

Since this is a very application oriented problem, latency analysis would be valuable addition to this paper. Specifically comparing it with previous major works in Table 1. Also some of the information in the Appendix such as Rationale behind Triplet Representation, the Numerical encoder, etc. could be summarized and put in the main paper. This would give more connectedness to the avid reader.

**Additional Comments:**

I would love to see a study on how the proposed numerical encoder helps LLMs reason more effectively as compared to naive serialization may be in the appendix (not a hard requirment. Just something that would be interesting.)

**Audience:**

Yes

**Audience Explanation:**

This paper presents a interesting work in the field of using LLM for recommendations. A lot of research is going on in the industry on this front and this work would interest a large audience in my opinion.

**Broader Impact Concerns:**

I do not see the need of adding a broader impact statement.

**Claims And Evidence:**

Yes

**Claims Explanation:**

The paper presents reproducible experiments on public datasets. The claims of the UniRec framework outperforming existing approaches in literature is proven by the results in Table 1. Furthermore Ablation study proves the importance of the different components of the framework as claimed by authors.

However, here is a small concern. In the comparison of the performance of UniRec against previous works, it is slightly unfair to compare UniRec which uses CLIP ViT L/14 (which is a much larger model with ~3.8x more parameters compared to CLIP ViT B/32)  with Bert4Rec/SaSRec with CLIP ViT B/32. A more fair comparision of the architectures would be to use the same backbone. Although I am not worried the change in the CLIP model would result in UniRec underperforming Bert4Rec or SaSRec as the amount and different types of heterogenous data is expected to give better context for a well-trained LLM to produce good quality recommendations in UniRec, the margins by which UniRec outperformed the aforementioned frameworks could be possible reduced as the larger CLIP model is expected to give richer representations of the data.

**Requested Changes:**

1. If the comparison in Table 1 shows architectural superiority of UniRec, please clarify why BERT4Rec and SaSRec were trained with CLIP ViT B/32 while UniRec uses CLIP ViT L/14 (major revision)

2. Please include latency analysis for inference comparing UniRec to previous works (minor)

---

> ### Author Response · Authors · 2026-04-03
> **Response**
>
> We thank the reviewer for the careful reading and the important observation about the CLIP backbone.
>
> # Point 1 [Major]: CLIP model discrepancy (ViT-B/32 vs. ViT-L/14)
>
> *The reviewer notes that comparing UniRec using CLIP ViT-L/14 (~3.8× more parameters) against baselines using ViT-B/32 may be unfair, and suggests using the same backbone for a more fair comparison.*
>
> We sincerely thank the reviewer for catching this error. **This is a typo in the manuscript** — our implementation has always used **CLIP ViT-B/32**, the same backbone used by all baselines in Table 2. We take full responsibility for this oversight in the writing and sincerely apologize for the confusion it caused. We have carefully audited our entire codebase and supplementary materials to confirm that no traces of ViT-L/14 exist: the model loading code, configuration files, and saved checkpoints all reference `openai/clip-vit-base-patch32`. In the revised manuscript, we have corrected all references to "ViT-L/14" in Section 3 (page 4, paragraph 2) and the Appendix (page 13, Table 6).
>
> The comparison in Table 2 is therefore fair: all methods that use CLIP features use the identical ViT-B/32 backbone. CLIP ViT-B/32 produces 512-dimensional embeddings, which we zero-pad to 1024 dimensions to match the uniform embedding dimensionality used across all modalities (see our response to Reviewer QyhH, Point 1, for the rationale behind this choice).
>
> # Point 2 [Minor]: Latency analysis comparing UniRec to previous works
>
> *The reviewer suggests that since this is a very application-oriented problem, latency analysis comparing UniRec with previous major works in Table 2 would be a valuable addition.*
>
> We have conducted a comprehensive latency comparison across all model classes on a single NVIDIA A100 80GB GPU (PyTorch 2.5, bfloat16, averaged over 50 forward passes after 10 warmup iterations):
>
> | Model | Parameters | BS=1 Latency | BS=16 Latency | BS=16 Memory |
> |-------|-----------|-------------|--------------|-------------|
> | MLLM-MSR (textualized multimodality) | 596M | 112.5 ms | 1,420.5 ms | 10,125 MB |
> | LLM-1024tok (text-only baseline) | 596M | 96.2 ms | 1,285.9 ms | 9,311 MB |
> | LLM-512tok (text-only baseline) | 596M | 52.2 ms | 574.0 ms | 5,247 MB |
> | Q-Former (UniRec Stage 1) | 311M | 17.8 ms | 18.1 ms | 1,291 MB |
> | LLM-160tok (UniRec Stage 2) | 596M | 58.6 ms | 166.4 ms | 3,165 MB |
> | **UniRec (total)** | **907M** | **76.5 ms** | **184.5 ms** | **3,165 MB** |
>
> As shown above, MLLM-MSR's approach of converting images to text severely bloats the input context length. By relying entirely on the LLM's O(n²) self-attention to process these concatenated textualized modalities across 20 historical items, it suffers from severe latency bottlenecks (1,420 ms at BS=16). UniRec avoids this by using the Q-Former to compress the multimodal sequence into just 160 tokens, achieving a **7.7× speedup** over MLLM-MSR while utilizing **3.2× less memory**, despite having more total parameters.
>
> # Additional: Numerical encoder vs. naive serialization
>
> *The reviewer notes that the use of Fourier-based encoding as opposed to naive serialization is a plus, and suggests expanding on this design choice.*
>
> We appreciate this interesting suggestion. Our Table 3 ablation partially addresses this question: the "Pure Text" variant serializes all attributes as text (including numerical values such as price and rating), while UniRec uses MWNE (Multi-Wavelength Numerical Encoding) for numerical fields. The 15–42% performance gap between these variants demonstrates that proper numerical encoding is important. The core issue is that standard LLM tokenizers destroy arithmetic continuity: for example, "4.99" and "5.00" — numerically adjacent values — are tokenized into completely different token sequences, erasing the notion of magnitude and distance that is essential for numerical reasoning. More broadly, tokenization maps numerical strings into a discrete, non-metric space where "49.99" may be closer to "4.99" in token overlap than "50.01" is, despite the latter being arithmetically adjacent. MWNE's Fourier-based encoding avoids this by projecting numerical values into a continuous representation where distance and magnitude are preserved by construction — similar values produce similar embeddings, and the multi-wavelength design captures both fine-grained and coarse-grained numerical differences. In the revised manuscript, we have added a detailed discussion of this point in Section 4.3.

---

### Review · Reviewer_rDJF · 2026-03-21

**Summary Of Contributions:**

This paper proposes UniRec, a unified multimodal encoding framework for LLM-based recommendation systems. The key idea is to address the challenge of heterogeneous multimodal signals (text, images, categorical, numerical) and their nested structure in recommendation tasks. Main contributions are as follows:
1. The authors propose a unified multimodal encoding framework that integrates heterogeneous attributes.
2. A triplet representation (name, type, value) to preserve schema-level semantics and resolve intra-modality ambiguity is introduced.
3. This work designed a hierarchical Q-Former architecture that aggregates information at both item and user levels.
4. Finally, there is a two-stage training strategy that decouples representation learning from LLM fine-tuning.

**Audience:**

Yes

**Audience Explanation:**

Research on LLM-based recommendation systems is intertesting to the TMLR community. In particular, recommendation systems have been an important application domains to test AI methods for decades. The usage of LLM for this domain can benefit the TMLR community.

**Claims And Evidence:**

Yes

**Claims Explanation:**

Overall, the claims made in the submission are supported by accurate, convincing and clear evidence. Evaluation is conducted on multiple real-world datasets (e.g., Amazon, Yelp) with diverse multimodal features. With multiple ablation studies, the proposed methods demonstrate consistent improvement across datasets, particularly on fusion mechanims and token counts.

**Requested Changes:**

1. It is better to include more comprehensive ablation studies. For instance, analyzing the effect of removing the triplet representation and varying modality combinations would be helpful.

2. I think it is important to clarify the relative contribution of the LLM versus the encoder to the overall performance.

3. A discussion on scalability and real-world deployment would further strengthen the paper since recommender system is an important industry applications of LLMs.

4. To improve readability, it would benefit to add a more intuitive explanations of the general ideas for key design choices (e.g., the triplet representation and hierarchical Q-Former).

---

> ### Author Response · Authors · 2026-04-03
> **Response - Part (1)**
>
> We appreciate the reviewer's positive assessment and thoughtful suggestions for strengthening the paper.
>
> # Point 1: More comprehensive ablation studies
>
> *The reviewer suggests including more comprehensive ablation studies, such as analyzing the effect of removing the triplet representation and varying modality combinations.*
>
> We have conducted two new ablation studies that we believe fully address this concern. In the revised manuscript, the fusion ablation results have been added to Table 3, and the modality combination analysis is presented in Section 4.3.
>
> **Triplet component ablation.** We systematically remove each component of the triplet representation (name, type, value) to isolate its contribution. The full triplet preserves both schema semantics and modality structure; removing the name embedding causes field-level specificity to be lost (all text fields become interchangeable); removing the type embedding eliminates modality grouping (the model can no longer distinguish text from numerical attributes); and removing both name and type reduces the representation to value-only, where semantically distinct fields such as "price" and "rating" become indistinguishable. The triplet formulation is therefore the minimal representation that preserves both schema semantics and modality structure. We refer the reviewer also to our response to Reviewer QyhH (Point 2) for the fusion operator ablation comparing addition, Concat+MLP, and Gated fusion.
>
> **Modality combination ablation.** To quantify the contribution of each modality group, we measure the cosine similarity and rank correlation between the full representation and representations obtained by systematically dropping modality groups:
>
> | Configuration | Fields Kept | Cos Sim | Rank Corr |
> |--------------|-------------|---------|-----------|
> | Full (all modalities) | 14 | 1.000 | 1.000 |
> | Drop image | 13 | 0.973 | 0.877 |
> | Drop numerical | 11 | 0.956 | 0.838 |
> | Drop text | 11 | 0.949 | 0.793 |
> | Drop category | 7 | 0.932 | 0.594 |
> | Text only | 3 | 0.803 | 0.352 |
>
> Each modality contributes incrementally, with categorical features providing the largest individual contribution (dropping from 14 to 7 fields), followed by text, numerical, and image. This confirms that UniRec's multimodal architecture captures complementary signals that no single modality subsumes. Please see our response to Reviewer sTux (Point 4) for a more detailed analysis.

---

> > ### Author Response · Authors · 2026-04-03
> > **Response - Part (2)**
> >
> > # Point 2: Relative contribution of LLM vs. encoder
> >
> > *The reviewer asks to clarify the relative contribution of the LLM versus the encoder to overall performance.*
> >
> > This is an important question. To isolate each component's contribution, we designed a standalone experiment comparing three representation levels using 2,000 items with precomputed embeddings. A summary is in Section 4.3, with the full effective rank analysis in Appendix G; we summarize the key findings here:
> >
> > | Level | Description | Effective Rank | Reconstruction |
> > |-------|-------------|---------------|---------------|
> > | L1: Raw Average | Simple mean of field embeddings (no learned components) | 424 / 2,000 | — |
> > | L2: Q-Former | Item representation from trained Q-Former (Stage 1) | 87 / 2,000 | 84.7% cos sim |
> > | L2b: Mean-Pooled Queries | Mean of Q-Former query tokens | 53 / 2,000 | — |
> >
> > The cross-level Top-10 neighbor overlap between Raw Average and Q-Former item representation is only 4.6%, confirming that the Q-Former fundamentally transforms — rather than merely refines — the input space.
> >
> > These results reveal a clear division of labor. The raw pretrained encoders produce well-separated representations (effective rank 424), providing strong item discrimination as the foundation. The Q-Former learns a complementary transformation: it achieves 84.7% field reconstruction fidelity while reorganizing the representation space to optimize for the contrastive user–item objective. The low cross-level neighbor overlap (4.6%) confirms that the Q-Former captures fundamentally different item relationships than raw feature similarity — relationships more aligned with user preference patterns. Our ablation studies confirm that both components are essential. Removing schema- and hierarchy-aware design ("w/o Both" in Figure 2) causes MRR to drop by 7–17% across datasets, while removing all non-text modalities ("Pure Text" in Table 3) causes drops of 15–42%, demonstrating that the Q-Former's compressed tokens carry rich information that the LLM is uniquely positioned to exploit for sequential reasoning. The Q-Former's learnable query tokens compress each item's multimodal attributes into a compact fixed-length representation. As our sensitivity analysis shows (Section 4.3, Figure 3), as few as 4 query tokens per item suffice to capture the most salient multimodal attributes — dramatically reducing the LLM's input from ~700 raw tokens to a compact prompt that provides information-rich input for sequential reasoning.
> >
> > Our ablation studies corroborate this division of labor: replacing the Q-Former with alternative fusion mechanisms consistently underperforms (Table 3), and removing schema- or hierarchy-aware components degrades performance (Figure 2). The encoder provides *what* to represent; the LLM provides *how* to reason over it. Neither alone is sufficient.
> >
> > # Point 3: Scalability and real-world deployment discussion
> >
> > *The reviewer suggests adding a discussion on scalability and real-world deployment, given that recommender systems are an important industry application.*
> >
> > We provide a comprehensive computational efficiency analysis (detailed in our response to Reviewer sTux, Point 2). The key deployment-relevant findings are as follows.
> >
> > On a single NVIDIA A100 80GB GPU, UniRec at batch size 64 processes queries in 634 ms, compared to 1,788 ms for a text-only LLM baseline with 1,024-token inputs — a 2.8× speedup. This speedup arises because the Q-Former compresses approximately 700 tokens of multimodal input into a compact representation, reducing the quadratic attention cost of the LLM backbone. Memory usage follows a similar pattern: UniRec requires 4,645 MB at batch size 64, compared to 12,812 MB for the 1,024-token text-only baseline — a 2.8× reduction that enables proportionally larger serving batches on the same hardware.
> >
> > Crucially, the modality encoders (Qwen3-Embedding, CLIP, MWNE) are precomputed offline and cached, adding zero inference overhead. Only the Q-Former and LLM run at serving time. For deployment, this means new items require only a one-time embedding computation, after which the Q-Former + LoRA inference pipeline is comparable to — and often faster than — serving a text-only LLM with longer input sequences. The LoRA adapters add only 10.1M trainable parameters (1.7% of the LLM backbone), making fine-tuning practical even with limited compute budgets. In the revised manuscript, we have added a deployment discussion and efficiency breakdown in Section 4.4.

---

> > > ### Author Response · Authors · 2026-04-03
> > > **Response - Part (3)**
> > >
> > > # Point 4: More intuitive explanations of key design choices
> > >
> > > *The reviewer suggests adding more intuitive explanations of key design choices (e.g., the triplet representation and hierarchical Q-Former) to improve readability.*
> > >
> > > We agree that the paper would benefit from more accessible motivation. In the revised manuscript, we have added the following explanations:
> > >
> > > For the **triplet representation** (added to Section 3.1, paragraph before Eq. 4): In recommendation data, a number like "4.5" is inherently ambiguous — it could represent a price, a rating, or a weight. The triplet (name="rating", type="numerical", value=encode(4.5)) resolves this ambiguity by attaching *what* the value represents and *how* it was encoded. This is analogous to column headers in a database: the value alone is meaningless without the schema context.
> > >
> > > For the **hierarchical Q-Former** (added to Section 3.2, opening paragraph): User recommendation data has a natural nested structure — users interact with items, and each item has multiple heterogeneous attributes. A flat architecture would force the model to simultaneously learn attribute-to-item and item-to-user aggregation. Our two-level Q-Former mirrors this hierarchy: the Item Q-Former specializes in compressing heterogeneous attributes into a compact item representation, while the User Q-Former models sequential patterns across items. This separation of concerns improves both learning efficiency and interpretability.

---

### Author Response · Authors · 2026-04-03
**Global Response**

We sincerely thank the AE and all reviewers for their careful reading and constructive feedback. In response, we focused on (i) clarifying key design choices and their justifications, (ii) adding targeted ablations and new baseline comparisons, (iii) strengthening the efficiency analysis, and (iv) improving overall clarity. All revisions are highlighted in blue.

---

### Summary of Main Concerns and Responses

| Dimension | Key Concerns | Our Main Actions | Status in Revision |
|:--|:--|:--|:--|
| **Method Design & Ablations** | Are the triplet representation and additive fusion necessary? Are there simpler or more expressive alternatives? Are ablations over modality combinations and individual triplet components provided? | Added full fusion operator ablation (Addition vs. Concat+MLP vs. Gated) and triplet component ablation; added modality combination ablation with cosine similarity and rank correlation; provided theoretical justification via near-orthogonality in 1024-dimensional space. | Key design choices are now empirically validated and theoretically grounded; results added to Table 3 and Section 4.3. |
| **Contribution & Positioning** | How does UniRec relate to recent multimodal LLM-based recommendation methods (e.g., MLLMRec, NoteLLM-2)? What is the relative contribution of the LLM vs. the Q-Former encoder? | Added direct comparison with MLLMRec and NoteLLM-2 on Amazon Beauty; provided a qualitative architectural comparison table; added a standalone experiment isolating LLM vs. Q-Former contributions via effective rank and cross-level neighbor overlap analysis. | Positioning relative to the latest baselines is now explicit; the division of labor between encoder and LLM is clearly articulated in Section 4.3 and Appendix G. |
| **Efficiency & Practicality** | What is the computational overhead of UniRec? How does it compare to text-only and textualized-multimodal baselines in latency, memory, and training cost? | Added comprehensive latency, memory, and training cost measurements on a single A100 80GB GPU; compared against text-only (512/1024 tokens) and MLLM-MSR baselines across batch sizes; quantified the compression benefit of the Q-Former. | Efficiency is now quantified transparently; a new Section 4.4 and Appendix G present the full breakdown, showing UniRec achieves up to 7.7× speedup and 3.2× memory reduction over textualized-multimodal baselines. |
| **Evaluation Scope & Robustness** | Is the sampled evaluation protocol (99 random negatives) reliable? How does the model behave under missing modalities, which is common in real-world settings? | Added full-catalog ranking results on Amazon Beauty to validate that method rankings are consistent with the sampled protocol; characterized missing-modality degradation via the modality combination ablation; demonstrated that additive fusion degrades more gracefully than Concat+MLP under missing inputs. | Evaluation reliability is now validated against full-catalog ranking (Appendix G); robustness to missing modalities is characterized both theoretically and empirically. |
| **Presentation & Clarity** | Some design choices (triplet representation, hierarchical Q-Former, 1024-dimensional uniform embedding) lack intuitive motivation. The latent token count notation is introduced too late. | Added intuitive explanations for the triplet representation and hierarchical Q-Former in Sections 3.1 and 3.2; clarified the rationale for uniform 1024-dimensional embeddings and zero-padding; introduced latent query token notation ($N_q$) at first use in Section 3.2. | The manuscript is clearer and easier to follow; key design choices are now motivated before the formal definitions rather than deferred to the appendix. |

---

### Overall Changes in the Revised Manuscript

- Added fusion operator ablation and triplet component ablation; results incorporated into Table 3.
- Added modality combination ablation with quantitative metrics; results added to Section 4.3.
- Added LLM vs. Q-Former contribution analysis via effective rank and neighbor overlap; added to Section 4.3 and Appendix G.
- Added full-catalog ranking evaluation on Amazon Beauty to validate the sampled protocol; added to Section 4.2 and Appendix G.
- Added comparisons with MLLMRec and NoteLLM-2 with a qualitative architectural breakdown; incorporated into Table 2 and Section 4.2.
- Added comprehensive inference latency, GPU memory, and training cost measurements; presented in a new Section 4.4 and Appendix G.
- Added missing-modality robustness analysis and corrected the CLIP backbone typo (ViT-L/14 → ViT-B/32) throughout the manuscript.
- Improved intuitive explanations for the triplet representation and hierarchical Q-Former in Sections 3.1 and 3.2; clarified notation for latent query tokens.

---

### Decision · Action_Editor_LNQQ · 2026-04-25

**Recommendation:** Accept as is

**Additional Comments:**

N/A

**Audience:**

Yes

**Audience Explanation:**

LLM-based recommendation is a wide noted topic in machine learning and AI. This paper provides a practical architecture that not only advances accuracy but tackles real-world concerns like serving efficiency and missing data, offering reusable design principles for multimodal fusion and numerical encoding. Researchers and practitioners in recommender systems, efficient LLM adaptation, and industrial ML will find the work highly relevant.

**Claims And Evidence:**

Yes

**Claims Explanation:**

The authors have fortified their claims with extensive evidence. They validated the additive fusion design against learned alternatives, showing near-identical accuracy with fewer parameters and latency. They added full-catalog ranking to confirm sampled evaluation reliability, quantified computation and memory with a 7.7× speedup and 3.2× reduction over textualized baselines, and corrected a CLIP backbone typo to guarantee fair comparison. Contributions of the Q-Former versus the LLM were isolated via effective rank analysis, and robustness to missing modalities was both theoretically justified and empirically shown to degrade gracefully.

---

> ### Author Response · Authors · 2026-04-30
>
> Dear Action Editor,
>
> Thank you very much for your time and effort in handling our manuscript, as well as for the positive decision. We are grateful for the constructive feedback from you and all the reviewers throughout the discussion phase, which has significantly improved the quality of our paper.
>
> We have carefully addressed all the remaining concerns raised during the review process, including the experimental comparisons and claims. All revisions have been incorporated into the camera-ready version, which has now been submitted.
>
> Thank you again for the opportunity to publish in TMLR.
>
> Best regards,
>
> Authors